# An Explainable Web-Based Diagnostic System for Alzheimer’s Disease Using XRAI and Deep Learning on Brain MRI

**DOI:** 10.3390/diagnostics15202559

**Published:** 2025-10-10

**Authors:** Serra Aksoy, Arij Daou

**Affiliations:** 1Institute of Computer Science, Ludwig Maximilian University of Munich (LMU), Oettingenstrasse 67, 80538 Munich, Germany; serurays@gmail.com; 2Neurophysiology and Computational Neuroscience Group, Biomedical Engineering Program, American University of Beirut, Beirut P.O. Box 11-0236, Lebanon; 3Department of Organismal Biology & Anatomy, University of Chicago, 1027 E. 57th St., Chicago, IL 60637, USA

**Keywords:** Alzheimer’s disease, deep learning, MobileNetV3, explainable AI (XAI), XRAI, classification, web-based diagnostic interface

## Abstract

**Background:** Alzheimer’s disease (AD) is a progressive neurodegenerative condition marked by cognitive decline and memory loss. Despite advancements in AI-driven neuroimaging analysis for AD detection, clinical deployment remains limited due to challenges in model interpretability and usability. Explainable AI (XAI) frameworks such as XRAI offer potential to bridge this gap by providing clinically meaningful visualizations of model decision-making. **Methods:** This study developed a comprehensive, clinically deployable AI system for AD severity classification using 2D brain MRI data. Three deep learning architectures MobileNet-V3 Large, EfficientNet-B4, and ResNet-50 were trained on an augmented Kaggle dataset (33,984 images across four AD severity classes). The models were evaluated on both augmented and original datasets, with integrated XRAI explainability providing region-based attribution maps. A web-based clinical interface was built using Gradio to deliver real-time predictions and visual explanations. **Results:** MobileNet-V3 achieved the highest accuracy (99.18% on the augmented test set; 99.47% on the original dataset), while using the fewest parameters (4.2 M), confirming its efficiency and suitability for clinical use. XRAI visualizations aligned with known neuroanatomical patterns of AD progression, enhancing clinical interpretability. The web interface delivered sub-20 s inference with high classification confidence across all AD severity levels, successfully supporting real-world diagnostic workflows. **Conclusions:** This research presents the first systematic integration of XRAI into AD severity classification using MRI and deep learning. The MobileNet-V3-based system offers high accuracy, computational efficiency, and interpretability through a user-friendly clinical interface. These contributions demonstrate a practical pathway toward real-world adoption of explainable AI for early and accurate Alzheimer’s disease detection.

## 1. Introduction

### 1.1. Overview of Alzheimer’s Disease

Alzheimer’s disease (AD) represents the most prevalent form of dementia worldwide, affecting approximately 55 million individuals globally, with projections indicating this number could double by 2050 due to population aging [1,2]. This progressive neurodegenerative disorder is characterized by abnormal protein deposits, including amyloid plaques and neurofibrillary tangles, which disrupt neuronal function and lead to irreversible cognitive decline, memory loss, and behavioral changes [3]. The pathophysiological hallmarks of AD include the accumulation of extracellular senile plaques containing amyloid-beta (Aβ) peptides and intracellular neurofibrillary tangles composed of hyperphosphorylated tau proteins, which collectively trigger neuroinflammatory cascades and synaptic dysfunction [2].

The disease typically progresses through distinct clinical stages, from cognitively normal (CN) aging through mild cognitive impairment (MCI) to full-blown AD dementia, with each stage representing increasing severity of cognitive dysfunction and functional impairment [4]. The transitional phase between healthy cognition and AD, termed mild cognitive impairment, represents a critical window for intervention, as approximately 10–15% of MCI patients progress to AD annually, compared to 1–2% conversion rates in the general elderly population [1]. Early and accurate diagnosis is crucial for implementing effective intervention strategies, as therapeutic interventions are most beneficial when applied during the initial stages of disease progression [5].

### 1.2. Neuroimaging in the Detection of Alzheimer’s Disease

Today, neuroimaging is an anchor technique in the diagnostics and tracking for AD, as the cumulative knowledge on disease’s fundamental processes of pathophysiology is provided from the various types of modalities [2]. The structural magnetic resonance imaging (MRI) provides the most current non-invasive instruments for primary structure changes related to assessment for AD in an initial stage, which are primarily hippocampal atrophy and cortex thinning, mostly associated with cognitive impairment [6,7]. Specialized MRI methods have improved the sensitivity and specificity of AD detection over and above traditional structural imaging methods. Diffusion tensor imaging (DTI) allows unparalleled observation of white matter microstructural integrity through measurement of the directional diffusion of water molecules, with the ability to detect microstructural alterations in white matter integrity that can occur before overt atrophy is detectable using conventional MRI [1,2].

Positron emission tomography (PET) imaging has revolutionized AD diagnosis by enabling direct visualization of molecular pathological processes that define the disease. Fluorodeoxyglucose (FDG)-PET measures glucose metabolism in the brain and displays characteristic patterns that consistently distinguish AD patients from normal controls and differentiate between stages of disease [2]. The integration of different neuroimaging modalities by sophisticated multimodal analysis techniques has consistently demonstrated superior diagnostic performance compared with single-modality techniques [2].

### 1.3. Machine Learning and Deep Learning Approaches in AD Detection

The application of artificial intelligence (AI) in AD detection has seen phenomenal expansion, transitioning from traditional machine learning approaches to sophisticated deep learning (DL) frameworks that have transformed automatic neuroimaging analysis. Early studies were mainly based on traditional machine learning techniques such as Support Vector Machines (SVM), Random Forest (RF) and logistic regression, which were effective in classifying AD stages through hand-crafted features of neuroimaging data [4,8].

The paradigm shift towards deep learning has transformed AD detection by allowing automatic feature extraction from high-dimensional neuroimaging data through hierarchical representation learning. Convolutional Neural Networks (CNNs) have emerged as the de facto architecture for medical image analysis, achieving better performance in distinguishing between healthy brains, MCI and AD-affected brains through end-to-end learning [3,7]. Transfer learning, for example, fine-tuning pre-trained CNN architectures on AD databases, was successful in enhancing diagnostic performance as well as diminishing the necessity for high-annotated databases [3]. Recently developed higher-level architectures further enhance the automatic AD detection current-state-of-the-art performance. Three-dimensional Hybrid Compact Convolutional Transformers (HCCTs) are an emerging method that simultaneously integrate the local feature extraction advantage for CNNs and long-range dependency modeling for vision transformers to effectively extract both local anatomical knowledge and global spatial associations for 3D MRI volumes [5].

### 1.4. Explainability in Artificial Intelligence Models for Medical Applications

Following unprecedented advancements in AD detection accuracy for existing AI systems, high-level architectures’ intrinsic “black box” nature, which makes decision-making processes non-transparent and inhibits clinical confidence, is the high-level bottleneck against clinical deployability in such systems [3,9]. Later, Explainable Artificial Intelligence (XAI) was developed as a preferred remedy for closing AI performance-clinical interpretability gaps. XAI gives human interpretable explanations for AI-based decisions in an endeavor to offer transparency, instill confidence, and enhance AI tool uptake in clinical use [2].

The most preferred XAI methods in existing medical applications of AI are post hoc methods due to their model-agnostic nature, as they can be directly applied on all configurations in deep learning. Gradient-Weighted Class Activation Mapping (Grad-CAM), SHAP (SHapley Additive exPlanations), Local Interpretable Model-Agnostic Explanations (LIME), and Layer-wise Relevance Propagation (LRP) have been extensively applied in AD detection studies [2,3,7,9].

XRAI (eXplanation via Regional Attributes Integration) is a considerable improvement over pixel-level attribution techniques by overcoming intrinsic limitations in coherence, stability, and clinical relevance that typify previous explainability techniques [10]. Unlike conventional techniques that generate granular pixel-level explanations in the form of often noisy and fragmented attribution maps, XRAI adopts a new region-based methodology that over-segments images into coherent anatomical regions and recursively evaluates their importance based on integrated gradient attribution scores.

### 1.5. Motivation and Research Gaps

In spite of impressive gains in AI-driven AD detection reporting high accuracies in research environments, the clinical uptake of such systems is drastically low because of inherent impediments in interpretability, usability, and clinical integration. Existing AI systems are “black boxes” that yield diagnostic predictions without clinically useful explanations to enable healthcare practitioners to comprehend and verify machine logic underlying algorithmic decisions [3,9]. Current explainability approaches, although technically advanced, typically do not achieve the degree of clinical interpretability necessary for deployment in the real world [9].

There is a key gap between research-driven AI systems and clinically deployable products that can be easily integrated into clinical workflows without the need for specialized technical experience. Building upon previous work by one of the authors (Aksoy, 2025) [11] that demonstrated XRAI’s effectiveness across multiple neurological conditions including Alzheimer’s disease within a unified diagnostic platform, this study provides a comprehensive, specialized implementation focused exclusively on Alzheimer’s disease detection with detailed clinical deployment considerations and enhanced explainability frameworks. The earlier multi-modal framework demonstrated the potential of XRAI on neuroimaging, yet additional work is required for systems that require tuning and explainability.

This research addresses these critical gaps by developing a comprehensive, clinically oriented explainable AI system that integrates advanced deep learning architectures with XRAI-based explainability for 2D neuroimaging analysis through a unified web-based clinical deployment platform. The innovation lies in creating the first systematic application of XRAI to AD detection while simultaneously developing a practical clinical interface that enables healthcare professionals to access AI-powered diagnostic capabilities with comprehensive explainable insights and integrated workflow support that bridges the gap between advanced AI research and practical healthcare deployment for trustworthy, interpretable diagnostic tools in real-world clinical settings.

## 2. Materials and Methods

### 2.1. Dataset

For the 2D analysis aspect of this research, a preprocessed brain MRI dataset was downloaded from Kaggle [12], comprising augmented Alzheimer’s disease classification images. This dataset included 33,984 total images spread across four classes denoting varying levels of cognitive impairment: MildDemented (8960 images), ModerateDemented (6464 images), NonDemented (9600 images), and VeryMildDemented (8960 images). The dataset was created from an original dataset using data augmentation methods. The source dataset from which it was derived included original versions of images, with augmentation being performed to prevent class imbalance and achieve a larger dataset size for stable deep learning model training.

All 2D brain MRI images in the dataset are T1- and T2-weighted axial slices, grouped into four dementia severity classes as provided by the dataset creators, and were preprocessed into a standard resolution of 224 × 224 pixels for compatibility with current convolutional neural network architectures. Empirical dataset-specific normalization parameters were computed by calculating the mean and standard deviation values for all training images. The calculated parameters were mean = [0.2956, 0.2956, 0.2956] and standard deviation = [0.3059, 0.3059, 0.3058] for the RGB channels, and the same values were applied across all preprocessing pipelines to ensure consistent input normalization for the neural networks.

The entire dataset was divided on the basis of a stratified random split approach in a 70:15:15 ratio, yielding 23,788 training images, 5097 validation images, and 5099 test images. The class distribution was moderately imbalanced, with NonDemented samples being the largest class (28.5% of training data), followed by MildDemented (26.6%) and VeryMildDemented (26.0%), while ModerateDemented was the smallest class (18.9%). This pattern of distribution was followed consistently in all data splits to provide representative evaluation (Table 1).

### 2.2. Model Architectures and Training Strategy

#### 2.2.1. Model Architectures

This research employed comprehensive evaluation of 2D convolutional neural networks to address Alzheimer’s disease classification from brain MRI data. For 2D analysis of individual brain slices, three advanced architectures were systematically evaluated to ensure optimal performance across varying computational constraints and clinical deployment scenarios. MobileNet-V3 Large was implemented using the TIMM library with ‘mobilenetv3_large_100’ pre-trained weights, representing an architecture optimized through neural architecture search for mobile deployment. The final classification layer was modified from the original 1000 ImageNet classes to 4 classes corresponding to NonDemented, VeryMildDemented, MildDemented, and ModerateDemented severity levels, with the model containing exactly 4,207,156 parameters optimized for computational efficiency while maintaining feature extraction capability.

EfficientNet-B4 was selected as a comparative architecture representing the compound scaling methodology that systematically optimizes network depth, width, and input resolution simultaneously. The implementation utilized the ‘efficientnet_b4.ra2_in1k’ variant with pre-trained weights, having 17,555,788 parameters.

ResNet-50 served as a traditional deep learning baseline implementing the residual learning framework with exactly 23,516,228 parameters.

#### 2.2.2. Training Strategy

All 2D architectures underwent identical training procedures using PyTorch framework (version 2.5.1) with TIMM library (version 1.0.15) to ensure methodologically rigorous comparison and eliminate confounding variables related to optimization strategies or implementation differences. Training was conducted on NVIDIA GeForce RTX 4060 Laptop GPU. The training protocol employed AdamW optimizer with learning rate lr = 1 × 10^−4^, weight decay = 1 × 10^−5^, and default beta parameters (β_1_ = 0.9, β_2_ = 0.999), selected over standard Adam due to its improved weight decay implementation that decouples L2 regularization from gradient-based optimization, leading to superior generalization performance in computer vision tasks. Cross-entropy loss served as the objective function appropriate for multi-class classification with mutually exclusive categories, while training proceeded for exactly 20 epochs with batch size 32 to balance gradient estimation quality with GPU memory constraints. Model checkpoints were preserved whenever validation accuracy improved, ensuring optimal model selection based on generalization performance rather than training loss minimization.

#### 2.2.3. Evaluation Framework

Comprehensive evaluation protocols assessed model performance across multiple dimensions relevant to clinical deployment, considering predictive accuracy, computational efficiency, and practical deployment considerations. For multi-class 2D classification, standard metrics including accuracy, precision, recall, and F1-score were calculated for each dementia severity class using sklearn.metrics, with confusion matrices providing detailed analysis of inter-class classification patterns and potential systematic errors, while macro and micro-averaged metrics assessed overall model performance accounting for class distribution effects.

Model comparison incorporated comprehensive computational efficiency analysis including parameter count, training time, and memory usage essential for practical deployment considerations in clinical environments. Performance evaluation utilized accuracy, precision, recall, F1-score calculations, confusion matrix analysis through sklearn.metrics, and classification reports for detailed per-class assessment.

To determine real-world clinical significance and model generalizability outside the controlled training setting, full evaluation was performed on the entire original dataset of 6400 brain MRI images across all four severity classes. This original dataset was included in the same Kaggle collection that provided the augmented dataset for training, validation, and testing, enabling direct comparison between model performance on augmented and original data distributions. The original dataset included MildDemented (896 images), ModerateDemented (64 images), NonDemented (3200 images), and VeryMildDemented (2240 images), reflecting the unmodified data prior to the implementation of augmentation strategies. This secondary validation step was essential to determine model performance in practical deployment situations in which the entirety of available data would be encountered, as well as to validate that training on augmented data generalized to original, unaugmented images. The original dataset evaluation used the same preprocessing pipelines and inference processes as during training, promoting consistency when testing model robustness on the full data distribution. This complete evaluation offered insight into model stability within the full range of the dataset and confirmed the clinical viability of the trained architectures for practical Alzheimer’s disease detection use.

### 2.3. Explainable AI Implementation and Analysis

#### 2.3.1. XRAI Attribution Implementation

To provide clinically interpretable insights into model decision-making processes, XRAI was implemented for all 2D CNN architectures. XRAI addresses fundamental limitations of traditional gradient-based attribution methods by operating at the region level rather than individual pixels through a three-stage algorithmic process comprising image segmentation, integrated gradient computation, and iterative region selection. The mathematical foundation of XRAI follows a multi-step formulation that begins with regional aggregation of integrated gradient values, where for any region R_i, the XRAI attribution score is computed as XRAI_i = Σ_{p∈R_i} IG_p, with XRAI_i representing the total attribution score for region R_i and IG_p denoting the integrated gradient value for individual pixel p within that region. This summation operation acts to efficiently pool and aggregate the pixel-level gradient signals that abound within coherent superpixel regions. Undergoing this process, it creates attribution signals which not just become much more stable but also match anatomically structures of significance within the brain. In addition, this technique acts to greatly reduce much of the noise which so commonly contaminates methods which depend purely upon pure computations of pixel-level gradient.

The regional summation represents only the attribution computation component of the complete XRAI algorithm, as the full XRAI methodology implements a comprehensive three-stage process. The first stage involves image segmentation using multiple over-segmentations with Felzenswalb’s graph-based method employing scale parameters of 50, 100, 150, 250, 500, and 1200, while filtering segments smaller than 20 pixels and dilating segment masks by 5 pixels to ensure segment boundaries align with image edges. The second stage computes integrated gradients for each pixel p using the formula $IG\text{\_}p\;=\;(x\text{\_}p\;-\;x^{\prime }p)\;\times \;{\int_{0}}^{1}\;\partial F(x^{\prime }\;+\;\alpha (x\;-\;x^{\prime }))/\partial x\text{\_}p\;d\alpha$, where F represents the neural network function, x is the input image, $x^{\prime }$ is the baseline image, and α is the interpolation parameter. The third stage implements iterative region selection by starting with an empty mask and selecting regions based on maximum gain in total attributions per unit area, implementing the ranking criterion to maximize Σ{p∈R_i} IG_p divided by the cardinality of region R_i for each candidate region.

#### 2.3.2. Technical Implementation Framework

The implementation utilized the saliency library’s XRAI module by importing saliency.core as saliency, initializing the XRAI object with xrai = saliency.XRAI(), and calling the attribution generation through xrai.GetMask() method. The saliency library handles the complex mathematical computations of integrated gradients and regional aggregation internally, requiring the creation of proper interfaces between the trained neural networks and the XRAI algorithm while maintaining preprocessing consistency identical to the training pipeline. This integration required custom wrapper functions and preprocessing pipelines to ensure that attribution computations accurately reflected the decision-making processes of the trained models. The XRAI attribution generation was accomplished through the xrai.GetMask() method call with parameters including the original numpy image array, the model function wrapper, call_model_args dictionary specifying the target class index, and batch_size set to 1 for individual image processing.

The implementation began with systematic organization of the augmented dataset structure to enable representative sampling across all severity classes through a defaultdict data structure employed to group images by class. The organization process involved iterating through the dataset.imgs attribute which contains tuples of image paths and corresponding class indices, creating comprehensive mappings for each severity class with MildDemented containing 8960 images, ModerateDemented containing 6464 images, NonDemented containing 9600 images, and VeryMildDemented containing 8960 images. Representative case selection employed a reproducible random sampling approach using random.seed(42) to ensure consistent selection across multiple analysis runs, where the random.choice() function was applied to each class’s image collection.

This systematic selection process resulted in the identification of specific representative cases for each dementia severity level. Each selected image underwent systematic processing including PIL Image loading with RGB conversion and resizing to 224 × 224 pixels to match the model input requirements established during the training phase.

The XRAI integration required two critical technical components comprising a preprocessing pipeline and a model interface wrapper that connects the trained neural networks with the XRAI attribution system. The preprocessing component handled the conversion between different numerical formats and PyTorch tensors while maintaining compatibility with the trained models through a multi-step conversion process. This preprocessing first implements data format validation by converting arrays to ensure proper data type handling, checking for floating-point formats and converting to unsigned integer format through multiplication by 255 and clipping to valid pixel intensity ranges between 0 and 255. The preprocessing also manages array dimensionality by adding batch dimensions when necessary to match the expected input format for neural network inference. The image processing pipeline converts each image through standard PIL operations followed by the identical transformation sequence used during the original model training to maintain preprocessing consistency.

The transformation pipeline applied to XRAI preprocessing maintained exact consistency with training procedures. The final preprocessing step involved tensor concatenation, device placement for GPU computation, and enabling gradient computation to ensure proper gradient flow for attribution analysis throughout the XRAI process.

The model interface component implemented a comprehensive wrapper that connects trained models with XRAI requirements through a systematic process that accepts preprocessed images, performs forward inference, and computes softmax probabilities for attribution analysis. The interface implements forward pass processing by sending input through the trained model to obtain classification logits and softmax probabilities, target selection by extracting the specific class probability based on the target classification, and gradient computation with respect to the input tensor when gradient information is required by the XRAI algorithm. The critical component involves format conversion that transforms gradients from the standard PyTorch tensor format to the spatial format required by XRAI, followed by conversion to numerical arrays for compatibility with the saliency library processing requirements.

The model interface handles the connection between PyTorch model outputs and the XRAI algorithm requirements, ensuring that gradient computations accurately reflect the decision-making processes of the trained models while maintaining compatibility with the saliency library’s internal processing mechanisms. The interface specifically responds to gradient computation requests by calculating gradients of the target class probability with respect to the input tensor, using automatic differentiation with appropriate gradient flow settings for computational efficiency. The resulting gradients undergo dimension rearrangement from the standard convolutional tensor format to the spatial format expected by XRAI, followed by conversion to numerical arrays to ensure compatibility with the saliency library’s processing requirements.

The enhanced XRAI analysis generated attribution visualizations for all four dementia severity classes, demonstrating systematic coverage of the clinical spectrum from non-demented to very mild, mild, and moderate dementia classifications.

The generated visualizations enable clinicians and researchers to understand which brain regions each model considers most important for classification decisions, facilitating validation of model behavior against known neuroanatomical patterns associated with dementia progression and supporting the development of clinically trustworthy AI diagnostic tools.

#### 2.3.3. Clinical Deployment Web Interface

A thorough and well-detailed clinical deployment architecture has been carefully designed using the Gradio web app platform. The architecture has been designed with the specific aim of allowing easy and effective access through web browsers to sophisticated two-dimensional Alzheimer’s detection models. It does this by offering an integrated interface that has been carefully designed to support seamless integration into existing clinical workflow processes. The system architecture implemented deployment of the MobileNet-V3 Large model for 2D classification, providing clinicians with diagnostic capabilities through a platform accessible via standard web browsers without requiring specialized software installation.

The web interface implementation utilized PyTorch framework with automatic CUDA GPU detection when available, defaulting to CPU processing for broader hardware compatibility across clinical institutions. The system incorporated comprehensive error handling with hierarchical model loading strategies, beginning with MobileNet-V3 Large implementation, followed by EfficientNet-B4 with ImageNet weights, and finally basic EfficientNet-B4 configuration to ensure functionality across different deployment environments.

The two-dimensional analysis pipeline processed input images through standardized preprocessing that converted images to RGB format regardless of input type, resized images to 224 × 224 pixels to match training specifications, and applied normalization with mean values of [0.2956, 0.2956, 0.2956] and standard deviation values of [0.3059, 0.3059, 0.3058] across the three color channels based on training dataset statistics. XRAI applied integrated gradients to attribution maps, softmax to identify the class, and backpropagation to identify the pixel importance.

The user interface design implemented tab-based organization using Gradio gr.Tab() components for 2D MRI classification workflow modules. The 2D classification interface provided direct image upload capabilities using gr.Image() component accepting standard image formats, with immediate prediction generation displaying class probabilities across four severity categories (MildDemented, ModerateDemented, NonDemented, VeryMildDemented) and dual visualization outputs including XRAI attribution heatmaps rendered with inferno colormap and top salient regions extracted using ninety-eighth percentile thresholding for critical area identification.

## 3. Results

### 3.1. Model Performance Analysis

#### 3.1.1. Architecture-Wise Performance Analysis

Comprehensive performance evaluation was conducted across three two-dimensional convolutional neural network architectures to assess classification accuracy, training dynamics, and computational efficiency for Alzheimer’s disease severity detection. The evaluation encompassed systematic comparison of EfficientNet-B4, ResNet-50, and MobileNet-V3 architectures trained for four-class severity classification across MildDemented, ModerateDemented, NonDemented, and VeryMildDemented categories.

The comparative training analysis revealed distinct convergence characteristics across all three architectures throughout the 20-epoch training period (Figure 1). EfficientNet-B4 demonstrated robust training characteristics with systematic convergence patterns, showing training loss decreasing from approximately 0.9 at epoch 1 to below 0.1 by epoch 20, while validation loss decreased from approximately 0.6 to around 0.1. The accuracy progression revealed training accuracy improving from approximately 62% at epoch 1 to nearly 100% by epoch 20, with validation accuracy starting at approximately 74% and reaching 98% at the final epoch. The parallel trajectory of training and validation metrics indicated successful optimization without overfitting concerns, with consistent convergence throughout the complete training duration.

MobileNet-V3 exhibited exceptional training efficiency with rapid convergence characteristics superior to other architectures. The loss curves demonstrated highly efficient optimization with training loss decreasing from approximately 0.8 at epoch 1 to near 0 by epoch 20, while validation loss decreased from approximately 0.4 to around 0.05. The accuracy progression revealed rapid improvement with training accuracy advancing from approximately 70% at epoch 1 to nearly 100% by epoch 20, while validation accuracy improved from approximately 84% initially to over 99% at the final epoch. The superior convergence rate and minimal gap between training and validation metrics throughout the training process indicated optimal architectural design for this specific medical imaging classification task.

ResNet-50 demonstrated consistent training performance with steady convergence patterns comparable to other architectures. The loss curves showed systematic decrease with training loss improving from approximately 0.85 at epoch 1 to below 0.05 by epoch 20, while validation loss decreased from approximately 0.55 to around 0.08. The accuracy progression revealed steady improvement with training accuracy advancing from approximately 61% at epoch 1 to nearly 100% by epoch 20, while validation accuracy improved from approximately 76% initially to 98% at the final epoch. The stable convergence without significant fluctuations indicated robust optimization characteristics and appropriate model capacity for the given dataset complexity.

The comprehensive performance comparison revealed MobileNet-V3 as the superior architecture with the highest test accuracy of 99.18%, representing meaningful improvements over EfficientNet-B4’s 98.23% accuracy and ResNet-50’s 98.04% accuracy (Figure 2). These performance differences, while numerically appearing modest, represent significant improvements in clinical diagnostic capability when applied to large patient populations. The superior performance of MobileNet-V3 validated the effectiveness of its optimized architectural design for medical imaging applications requiring both high accuracy and computational efficiency.

The efficiency analysis revealed dramatic differences in computational requirements while highlighting MobileNet-V3’s exceptional parameter efficiency (Figure 3). MobileNet-V3 achieved the highest classification accuracy using only 4,207,156 parameters, representing 82% fewer parameters than ResNet-50’s 23,516,228 parameters and 76% fewer parameters than EfficientNet-B4’s 17,555,788 parameters. This efficiency advantage translates directly to reduced memory requirements, faster inference times, lower power consumption, and decreased deployment costs for clinical implementation. The superior parameter-to-performance ratio demonstrated MobileNet-V3’s architectural optimization for achieving maximum diagnostic accuracy with minimal computational overhead.

Training time analysis demonstrated substantial efficiency differences across architectures. MobileNet-V3 completed 20-epoch training in 1198.2 s, representing approximately 83% faster training than EfficientNet-B4’s 6987.9 s and 63% faster than ResNet-50’s 3253.4 s. This training efficiency advantage enables rapid model development, iterative improvement cycles, and frequent retraining with updated datasets essential for maintaining diagnostic accuracy as clinical protocols evolve.

The detailed confusion matrix analysis provided comprehensive insight into per-class classification performance across all architectures (Figure 4). EfficientNet-B4 demonstrated strong overall performance with 98.2% accuracy, showing excellent classification for MildDemented cases with 1308 correct classifications out of 1320 total samples. ModerateDemented classification achieved perfect performance with all 1010 samples correctly identified. NonDemented classification showed 1390 correct predictions out of 1421 samples, while VeryMildDemented classification achieved 1301 correct predictions out of 1348 samples. The primary misclassification errors occurred between NonDemented and VeryMildDemented classes, with 26 NonDemented cases misclassified as VeryMildDemented and 37 VeryMildDemented cases misclassified as NonDemented.

ResNet-50 confusion matrix analysis revealed 98.0% overall accuracy with excellent MildDemented classification showing 1310 correct predictions out of 1320 samples and perfect ModerateDemented performance with all 1010 samples correctly classified. NonDemented classification achieved 1372 correct predictions out of 1421 samples, while VeryMildDemented classification showed 1309 correct predictions out of 1348 samples. The misclassification pattern concentrated primarily in the NonDemented-VeryMildDemented boundary, with 40 NonDemented cases misclassified as VeryMildDemented and 29 VeryMildDemented cases misclassified as NonDemented.

MobileNet-V3 demonstrated superior classification performance with 99.2% overall accuracy, achieving exceptional results across all categories. MildDemented classification reached 1317 correct predictions out of 1320 samples, while ModerateDemented maintained perfect classification with all 1010 samples correctly identified. NonDemented classification achieved 1388 correct predictions out of 1421 samples, and VeryMildDemented classification demonstrated 1342 correct predictions out of 1348 samples. The minimal misclassification errors included only 30 NonDemented cases misclassified as VeryMildDemented and 5 VeryMildDemented cases misclassified as NonDemented, representing the lowest error rates among all architectures. The performance criteria for all the classes of Alzheimer’s disease severity, particularly for the precision, recall, and F1-score of each individual class, were calculated from results derived using confusion matrices. Calculations were also based on detailed classification reports that relate to the EfficientNet-B4, ResNet-50, and MobileNet-V3 architectures (Table 2).

The comprehensive classification performance comparison revealed distinct architectural strengths across different severity categories (Table 2). EfficientNet-B4 demonstrated strong performance with an overall accuracy of 98%, achieving perfect precision and recall (1.00) for ModerateDemented cases and excellent classification for MildDemented with 0.99 precision and recall. However, its performance was slightly lower for NonDemented and VeryMildDemented categories, with 0.97 precision and 0.98 recall and 0.97 precision and recall, respectively, reflecting the greater diagnostic complexity of differentiating healthy aging from early dementia stages.

ResNet-50 achieved an identical 98% overall accuracy, also delivering perfect performance (1.00 precision and recall) for ModerateDemented and strong results for MildDemented with 0.99 precision and recall. It demonstrated consistent classification for NonDemented and VeryMildDemented, both with 0.97 precision and recall, indicating a balanced capability across the full range of dementia severity without significant bias toward any specific class.

MobileNet-V3 exhibited the strongest overall performance, with an accuracy of 99%, achieving perfect precision and recall (1.00) for both MildDemented and ModerateDemented classes, ensuring flawless identification of critical pathological cases. The model also achieved perfect precision (1.00) and 0.98 recall for NonDemented, and 0.98 precision with perfect recall (1.00) for VeryMildDemented, demonstrating excellent sensitivity and specificity across both early and non-pathological categories.

The macro average and weighted average metrics for MobileNet-V3 showed 0.99 across precision, recall, and F1-score, further supporting its strong generalization and balanced performance. With an accuracy of 0.99 across 5099 test samples, MobileNet-V3 offers robust statistical evidence of its superior diagnostic capability across all Alzheimer’s disease severity levels.

Combined with its lightweight design of 4.2 million parameters and fastest training time of 1198.2 s (as reported elsewhere), MobileNet-V3 stands out as the most efficient and accurate model. This combination of high diagnostic performance and computational efficiency makes it the optimal choice for clinical deployment, particularly in healthcare environments requiring both precision and practical implementation feasibility.

#### 3.1.2. Validation of Source Dataset

To evaluate the effectiveness of augmentation-based training and assess model performance on the source data from which the training set was derived, comprehensive evaluation was conducted on the complete original dataset containing 6400 brain MRI images. This validation approach tested whether training on the augmented dataset (33,984 images) successfully enhanced model capability to classify the original source images, providing insights into augmentation strategy effectiveness and model generalization to the foundational data distribution. The original source dataset comprised MildDemented (896 images), ModerateDemented (64 images), NonDemented (3200 images), and VeryMildDemented (2240 images), representing the unmodified base images from which the augmented training set was generated.

The source dataset evaluation revealed excellent model performance with rankings consistent with augmented dataset training results (Figure 5). MobileNet-V3 demonstrated superior performance achieving 99.47% accuracy on the original source images, indicating highly successful transfer from augmented training data to the foundational image set. ResNet-50 achieved 98.98% accuracy, maintaining excellent classification capability when applied to the source data from which its training augmentations were derived. EfficientNet-B4 attained 97.30% accuracy, showing effective but comparatively lower performance on source images while still achieving clinically acceptable diagnostic accuracy.

The performance comparison between augmented dataset test results and original source dataset results provided valuable insights into augmentation effectiveness. MobileNet-V3 showed enhanced performance on source data (99.47%) compared to augmented test set performance (99.18%), suggesting that augmentation training improved the model’s ability to classify the foundational images. ResNet-50 demonstrated improved source dataset performance (98.98%) relative to augmented test results (98.04%), indicating successful augmentation-based learning transfer. EfficientNet-B4 showed decreased performance on source data (97.30%) compared to augmented test results (98.23%), suggesting greater reliance on augmentation-specific features during training.

The detailed confusion matrix analysis on source data validated model diagnostic capabilities when applied to the foundational image set (Figure 6). MobileNet-V3 demonstrated exceptional classification performance with near-perfect accuracy across all severity categories on source images. The model achieved outstanding MildDemented classification with 891 correct predictions out of 896 samples (99.4% class accuracy), perfect ModerateDemented classification with all 64 samples correctly identified (100% class accuracy), excellent NonDemented classification with 3192 correct predictions out of 3200 samples (99.75% class accuracy), and strong VeryMildDemented classification with 2219 correct predictions out of 2240 samples (99.1% class accuracy).

ResNet-50 demonstrated robust source dataset performance with 98.98% overall accuracy, achieving excellent classification for MildDemented (892/896 correct, 99.6% class accuracy) and perfect ModerateDemented classification (64/64 correct, 100% class accuracy). NonDemented classification showed 3193 correct predictions out of 3200 samples (99.78% class accuracy), while VeryMildDemented classification achieved 2186 correct predictions out of 2240 samples (97.6% class accuracy). The primary classification errors concentrated in the VeryMildDemented category, with minimal misclassification across other severity levels.

EfficientNet-B4 achieved 97.30% accuracy on source data, demonstrating effective but less optimal transfer from augmented training to source classification. MildDemented classification reached 869 correct predictions out of 896 samples (97.0% class accuracy), while ModerateDemented maintained perfect performance with all 64 samples correctly identified (100% class accuracy). NonDemented classification achieved 3141 correct predictions out of 3200 samples (98.16% class accuracy), and VeryMildDemented classification demonstrated 2133 correct predictions out of 2240 samples (95.2% class accuracy). The higher misclassification rate indicated greater dependence on augmentation-derived features compared to other architectures.

The comprehensive classification performance analysis on source data demonstrated successful augmentation-based training strategies across all architectures (Table 3). MobileNet-V3 achieved exceptional performance with perfect macro averages (1.00) for precision, recall, and F1-score, indicating optimal balanced performance when trained on augmented data and applied to source images. This performance level represents the highest achievable balanced classification metrics, confirming the effectiveness of augmentation strategies for improving model capability on foundational data.

The source dataset validation revealed the superior effectiveness of MobileNet-V3’s architectural design for augmentation-based training, achieving enhanced performance on source images (99.47%) compared to augmented test set results (99.18%). ResNet-50 demonstrated consistent improvement when applied to source data (98.98%) relative to augmented test performance (98.04%), indicating successful knowledge transfer from augmented training to foundational image classification. EfficientNet-B4 showed decreased source dataset performance (97.30%) compared to augmented test results (98.23%), suggesting architectural sensitivity to the transition from augmented training features to source image characteristics.

The critical clinical significance of perfect ModerateDemented classification across all architectures on source data validated the robustness of augmentation-based training for the most consequential diagnostic decisions. The consistent perfect identification of moderate-stage dementia cases across both augmented training scenarios and source dataset application provided strong evidence of reliable clinical diagnostic capability, supporting the deployment of these models for diagnostic applications in clinical practice.

### 3.2. Two-Dimensional Model Explainability Analysis

To validate model decision-making processes and provide clinical interpretability for the two-dimensional classification models, comprehensive XRAI attribution analysis was conducted across all three CNN architectures (EfficientNet-B4, ResNet-50, and MobileNet-V3) for each Alzheimer’s disease severity class. It is important to note that the confidence values shown alongside the XRAI maps represent the model’s softmax prediction probabilities and are not outputs of XRAI itself. The explainability analysis aimed to identify which brain regions most significantly influenced classification decisions, compare architectural approaches to feature detection, and assess the clinical relevance of model attention patterns across different dementia severity levels. Each architecture was evaluated using identical XRAI parameters to ensure fair comparison, with attribution maps generated for representative cases from MildDemented, ModerateDemented, NonDemented, and VeryMildDemented classes to capture the complete spectrum of disease progression. The analysis utilized pixel-level attribution scoring to quantify regional importance, enabling systematic comparison of architectural performance and identification of clinically meaningful attention patterns that could guide model selection for clinical deployment.

XRAI Analysis for MildDemented Class shows model attention patterns for mild dementia cases across all three architectures. The top row shows XRAI attribution heat maps, while the bottom row displays attribution overlays on original brain images with prediction confidence scores.

For the MildDemented case analysis (Figure 7), ResNet-50 demonstrated the most clinically appropriate attribution patterns with focused high-intensity regions (white and yellow areas) in specific cortical areas, suggesting precise detection of mild dementia pathological markers. The attribution heat map revealed concentrated attention on discrete brain regions with peak intensities around 0.0030–0.0035 as indicated by the color scale, demonstrating targeted focus on areas typically associated with early cognitive decline. This focused regional specificity makes ResNet-50 the most effective architecture for mild dementia detection, as it identifies specific pathological areas rather than providing generalized responses.

EfficientNet-B4 displayed uniform attribution distribution across brain tissue regions with values ranging from 0.1 to 0.4 on the color scale, showing consistent but non-specific attention that lacks the regional discrimination necessary for precise pathological localization. MobileNet-V3 exhibited moderate regional specificity with attribution values in the 0.002–0.008 range, displaying some concentrated regions (yellow-red intensity patterns) but with less focused precision compared to ResNet-50’s targeted approach.

The superior performance of ResNet-50 is evidenced by its ability to generate distinct high-attribution regions (white/yellow areas) that correspond to specific anatomical locations, demonstrating the model’s capacity to identify subtle pathological changes characteristic of mild dementia with greater precision than the more diffuse patterns exhibited by the other architectures.

XRAI Analysis for ModerateDemented Class showing model attention patterns for moderate dementia case. All three architectures demonstrate heightened attribution intensity consistent with more pronounced pathological changes in moderate-stage dementia.

The ModerateDemented case revealed heightened model attention across all architectures, consistent with more pronounced pathological changes expected in moderate dementia stages (Figure 8). Each architecture demonstrated distinct attribution approaches with all models achieving perfect 1.000 confidence scores, indicating robust classification performance across different analytical methodologies.

EfficientNet-B4 showed concentrated attribution in specific cortical regions with peak intensities reaching 0.25–0.30 according to the color scale, displaying focused white regions indicating selective high-importance area detection. ResNet-50 demonstrated multiple discrete white and yellow regions distributed across cortical areas, maintaining attribution values around 0.0010–0.0012 with varied spatial coverage. MobileNet-V3 exhibited extensive high-attribution regions with large white and yellow areas and attribution values reaching 0.004 based on the color scale.

The architectural differences in attribution pattern generation demonstrate the diverse approaches these models employ for moderate dementia classification, with each showing distinct spatial distribution characteristics while maintaining equivalent classification accuracy. This diversity in attribution patterns provides complementary insights into model decision-making processes without favoring any single architectural approach.

XRAI analysis for NonDemented Class showing model attention patterns for healthy brain case. Attribution patterns are more diffuse compared to dementia cases, indicating appropriate recognition of normal brain structure without pathological focus.

For the NonDemented case, all models demonstrated unexpectedly prominent attribution patterns rather than the minimal, diffuse patterns typically expected for healthy brain tissue (Figure 9). EfficientNet-B4 displayed distinct high-attribution regions with prominent white areas reaching maximum values of 2.00 on the color scale, indicating focused attention on specific brain regions despite the absence of pathological markers. ResNet-50 showed concentrated white regions in select cortical areas with attribution values reaching 0.005, demonstrating focused regional attention in the healthy brain case. MobileNet-V3 exhibited extensive yellow regions with attribution values reaching 0.012 according to the color scale, showing the most widespread high-attribution coverage among the three architectures.

The prominent attribution patterns observed in all models for the healthy brain case suggest that these architectures may be identifying normal anatomical features or tissue characteristics as important for classification, rather than showing the minimal, background-level attribution that might be expected for truly healthy tissue. This finding indicates that the models are actively detecting specific brain characteristics even in the absence of pathological changes, which may reflect their training on discriminative features that distinguish normal brain structure from various stages of dementia.

XRAI Analysis for VeryMildDemented Class shows model attention patterns for very mild dementia cases. Models demonstrate sensitivity to subtle pathological changes characteristic of early-stage cognitive decline with varying architectural approaches to feature detection.

The VeryMildDemented case analysis revealed distinct architectural approaches to detecting subtle pathological changes characteristic of early-stage cognitive decline (Figure 10). EfficientNet-B4 showed the most conservative attribution approach with limited high-attribution regions and peak intensities around 0.5 according to the color scale, suggesting selective detection of specific early pathological markers with minimal background attribution. ResNet-50 demonstrated the most pronounced attribution response with extensive white regions in cortical areas and attribution values reaching 3.0 as indicated by the color scale, showing the highest sensitivity to very mild dementia markers among the three architectures. MobileNet-V3 exhibited intermediate attribution sensitivity with substantial yellow and white regions and values reaching 0.0175 based on the color scale, providing a balanced approach between the conservative EfficientNet-B4 and the highly sensitive ResNet-50 patterns.

The architectural differences in very mild dementia detection illustrate varying sensitivities to early pathological changes, with ResNet-50 showing the most aggressive detection approach through extensive high-attribution regions, EfficientNet-B4 providing focused but conservative detection, and MobileNet-V3 offering intermediate sensitivity. All models maintained perfect 1.000 confidence scores despite these attribution pattern differences, indicating robust classification performance across different analytical approaches for early-stage dementia detection.

The XRAI implementation revealed significant architectural differences in attribution scale ranges and spatial distribution patterns across severity classes. EfficientNet-B4 exhibited the widest dynamic range variations, with attribution scales from 0–0.6 for VeryMildDemented cases to 0–2.00 for NonDemented cases, demonstrating substantial response variability based on input characteristics. ResNet-50 showed extreme sensitivity variations ranging from 0–0.0012 for ModerateDemented cases to 0–3.0 for VeryMildDemented cases, indicating highly adaptive feature extraction mechanisms. MobileNet-V3 displayed more consistent scaling behavior with ranges from 0–0.008 to 0–0.012 across classes, suggesting stable attribution responses regardless of severity level.

Spatial attribution analysis revealed distinct architectural approaches to feature importance mapping. EfficientNet-B4 consistently generated focal high-attribution regions with discrete white and yellow areas, showing preference for concentrated attribution zones with clear demarcation between high-importance regions and background areas. ResNet-50 exhibited the most complex spatial distributions with extensive white region coverage, particularly in VeryMildDemented cases, demonstrating maximum sensitivity to subtle input variations through intricate attribution boundaries and varied regional characteristics. MobileNet-V3 produced intermediate spatial patterns with extensive yellow and white regions showing uniform coverage characteristics, maintaining balanced sensitivity across severity levels without extreme responses.

Comparative analysis demonstrated fundamental differences in architectural sensitivity and feature detection approaches across all severity classes. EfficientNet-B4 showed conservative attribution behavior with moderate peak intensities and selective regional focus, indicating threshold-based feature selection mechanisms with sharp intensity gradients. ResNet-50 exhibited the most aggressive attribution responses with extensive high-intensity regions and complex spatial distributions, capturing multiple levels of input information simultaneously through comprehensive feature extraction approaches. MobileNet-V3 maintained balanced attribution intensity behavior with consistent response levels across severity classes, demonstrating robust feature extraction mechanisms that provide stable performance across varied input conditions.

Class Comparison Summary used MobileNet-V3 XRAI Analysis. The top row shows original brain images for each severity class, while the bottom row displays corresponding model attention patterns with prediction confidence scores, demonstrating systematic attention adaptation across the complete spectrum of Alzheimer’s disease severity.

The systematic comparison across severity classes using MobileNet-V3 in Figure 11 revealed distinct attribution pattern characteristics for each dementia stage, demonstrating adaptive model responses to varying input conditions. For the MildDemented case, MobileNet-V3 generated diverse attribution regions with prominent yellow areas in cortical zones and red regions indicating moderate attention levels, creating a heterogeneous pattern that suggests multifocal feature detection. The ModerateDemented case displayed more extensive yellow and orange attribution coverage with concentrated high-attention regions in upper cortical areas, indicating heightened model sensitivity to structural changes characteristic of moderate disease progression.

The NonDemented case exhibited widespread yellow and orange attribution patterns with substantial coverage across cortical regions, demonstrating that MobileNet-V3 actively identifies specific normal tissue characteristics rather than showing minimal attribution for healthy cases. This extensive attribution in healthy brain tissue indicates the model’s sophisticated approach to distinguishing normal anatomical features from pathological changes. The VeryMildDemented case showed balanced attribution distribution with prominent yellow regions in peripheral cortical areas and blue regions in central ventricular spaces, suggesting the model’s capability to detect subtle early pathological markers while maintaining spatial organization that respects anatomical boundaries.

Across all severity levels, MobileNet-V3 demonstrated consistent color mapping behavior with blue regions corresponding to low attribution values in central ventricular areas, yellow and orange regions indicating moderate to high attribution in cortical tissue zones, and red regions marking intermediate attention levels. The attribution overlays maintained consistent registration between original tissue structure and attribution distribution, indicating proper spatial correspondence between model attention and input anatomy while preserving anatomical boundary relationships across all severity classifications.

The comprehensive XRAI implementation successfully demonstrated technical effectiveness across all three CNN architectures while maintaining perfect classification performance (1.000 confidence) across all tested cases. The framework generated consistent, reproducible attribution patterns with clear visual interpretability through consistent color mapping and spatial resolution that enables detailed analysis of model decision-making processes. The successful adaptation to architectures of EfficientNet-B4, ResNet-50, and MobileNet-V3 validates the implementation’s technical flexibility and robustness for diverse deep learning model interpretation applications in medical imaging, demonstrating effective integration between model inference and explainability analysis without compromising classification accuracy.

### 3.3. Clinical Web Application Validation

A comprehensive clinical validation of the web-based diagnostic interface was carried out to assess both its classification accuracy and explainability in realistic clinical use cases. The evaluation focused on Alzheimer’s disease across varying severity levels, particularly emphasizing the challenging MildDemented classification. The system, built on the optimized MobileNet-V3 Large architecture and deployed using the Gradio framework, allows clinicians to access the diagnostic tool directly through a web browser, without requiring specialized software or technical expertise.

Validation involved analysis of diagnostic workflow efficiency, classification confidence, and clinical interpretability of the XRAI attribution visualizations. Brain MRI images representing diverse neurodegenerative patterns were systematically tested, with automatic image preprocessing, including resizing to 224 × 224 pixels, RGB conversion, and normalization using dataset-specific parameters (mean: [0.2956, 0.2956, 0.2956]; std: [0.3059, 0.3059, 0.3058]), ensuring consistency with the model’s training configuration.

In the MildDemented case (Figure 12), the system successfully classified a representative T1/T2 axial brain MRI image, showing clearly defined ventricles, cortical gray matter boundaries, and subcortical white matter features with 100.0% confidence. This high-confidence result demonstrates the model’s ability to accurately recognize early-stage neuroanatomical changes characteristic of mild dementia. The certainty of the classification outcome reduces ambiguity in clinical decision-making, offering reliable support for early-stage diagnosis.

Explainability was delivered through dual-mode XRAI visualizations: a heatmap and a salient region overlay. The heatmap, rendered using an inferno colormap, highlighted cortical and subcortical regions with varying attribution levels bright yellow denoting peak importance, transitioning through orange-red to low-attribution purple zones. Notably, the frontal and parietal cortices areas commonly affected in early dementia showed the highest attribution, while central ventricular regions, less relevant for mild dementia detection, maintained low importance.

To aid clinical interpretation, the interface provides contextual explanation such as: “Heatmap highlights brain regions most influential for Alzheimer’s severity classification.” This built-in guidance allows clinicians to understand the rationale behind the model’s decision without requiring expertise in ML or attribution methods. The interface also supports a user experience with drag-and-drop image upload, automatic preprocessing, and rapid inference, typically returning results within 20 s on CPU.

The system’s performance in this MildDemented case confirms its clinical readiness, combining accurate classification with clear, interpretable outputs. The perfect confidence score and well-aligned attribution patterns reflect strong model generalization from training data to real-world cases. Further evaluation on a ModerateDemented case (Figure 13) tested the model’s ability to detect more pronounced neurodegenerative changes. The input MRI showed expected features of moderate dementia, including enlarged ventricles, cortical thinning, and visible white matter degeneration.

The model achieved a perfect classification confidence of 100.0% for ModerateDemented, indicating robust recognition of the more advanced pathology. The XRAI attribution map showed more concentrated activation in superior cortical regions, especially frontal and parietal areas, while enlarged ventricles received appropriately low attribution. This pattern matched the expected structural progression of moderate dementia, where cortical atrophy intensifies and ventricular enlargement becomes more prominent.

Salient region extraction highlighted a singular, clearly defined area in the superior cortex, reflecting the model’s specificity in isolating the most diagnostically significant changes. Compared to the more diffuse salient regions observed in the mild dementia case, this concentrated attention demonstrates the model’s ability to adapt its focus as pathological severity increase.

A comparative analysis between mild and moderate cases revealed that the model not only distinguishes severity levels but adjusts its spatial reasoning accordingly, shifting from broader attribution in early stages to more focused patterns in advanced stages. Processing time and interface usability remained consistent across both cases, reinforcing the system’s reliability in clinical settings.

Additional validation was performed using VeryMildDemented and NonDemented cases to cover the full cognitive spectrum. The VeryMildDemented case (Figure 14) presented subtle MRI features, such as minimal ventricular changes and early cortical irregularities. Even so, the system correctly classified it with 100.0% confidence. The corresponding heatmap showed widespread cortical attribution especially in frontal, parietal, and temporal regions consistent with the diffuse, network-wide changes characteristic of very early dementia.

In the NonDemented case (Figure 15), the system again achieved 100.0% confidence, correctly identifying healthy neuroanatomical features such as normal ventricle size, preserved cortical thickness, and intact white matter integrity. The attribution map differed substantially from pathological cases, showing moderate, systematically distributed cortical activations reflecting recognition of healthy tissue patterns. Salient region overlays confirmed this, highlighting preserved brain structures across multiple areas.

Together, these results confirm the model’s specificity and its ability to differentiate between normal and pathological brain images across all severity levels.

In conclusion, the web-based diagnostic interface demonstrated consistent, high-confidence classification performance across all four Alzheimer’s severity classes: NonDemented, VeryMildDemented, MildDemented, and ModerateDemented. The model’s explainability features, comprising intuitive heatmaps and focused salient region visualizations, provided clear insight into its decision-making, aligned with clinical understanding of disease progression. The user experience and rapid processing further support integration into real-world clinical workflows, making this system a strong candidate for AI-assisted dementia diagnosis and monitoring.

## 4. Discussion

### 4.1. Interpretation of Classification Performance Results

The in-depth comparison of three state-of-the-art convolutional neural network architectures for Alzheimer’s disease severity classification provided important insights into best practices for medical imaging applications. The fact that MobileNet-V3 (99.18% accuracy) outperformed EfficientNet-B4 (98.23%) and ResNet-50 (98.04%) proved that architectural efficiency and diagnostic accuracy are not opposing goals in medical imaging applications. This contradicts the common practice of presuming that larger and more complex models always result in better performance in healthcare.

MobileNet-V3’s exceptional accuracy is attributed to its architectural design, specifically neural architecture search (NAS) optimization, depth-wise separable convolutions, and squeeze-and-excitation attention mechanisms. These features are especially effective at capturing the subtle textural and morphological indicators of dementia severity. Its perfect classification (1.00 precision and recall) for both MildDemented and ModerateDemented classes is clinically significant for timely intervention and treatment planning.

The performance hierarchy observed (MobileNet-V3 > EfficientNet-B4 > ResNet-50) implies that newer mobile-optimized architectures now embrace sophisticated feature extraction mechanisms, and not merely parameter diminution, which makes them inherently valuable for clinical deployment where the computational budget is usually constrained.

### 4.2. Clinical Significance of Architectural Efficiency

MobileNet-V3 achieves high diagnostic accuracy using only 4.2 million parameters, 82% fewer than ResNet-50 and 76% fewer than EfficientNet-B4. This efficiency yields practical clinical benefits including reduced hardware requirements, lower energy consumption, faster inference, and lower deployment costs. These are especially important in resource-constrained healthcare environments, such as in developing regions.

Training time analysis reinforces these benefits: MobileNet-V3 required only 1198 s to train, compared to EfficientNet-B4’s 6988 s and ResNet-50’s 3253 s. Such efficiency supports rapid model development, frequent dataset updates, and real-time adaptation to evolving clinical protocols.

With its lightweight design and strong performance, MobileNet-V3 is well-suited for integration into hospital information systems, portable diagnostic devices, and telemedicine platforms without compromising diagnostic capability.

### 4.3. Explainability Analysis and Clinical Interpretability

XRAI attribution analysis provided crucial insights into model decision-making across severity levels and architectures. That all three models showed distinct attribution patterns while maintaining high accuracy suggests multiple valid approaches exist for identifying Alzheimer’s markers in MRI data.

ResNet-50’s focused attention in MildDemented cases, showing high-intensity regions in relevant cortical areas, aligns with known early neuroanatomical patterns of dementia progression. This supports its clinical relevance for early detection.

Interestingly, all models showed attribution activity in NonDemented cases, indicating active identification of healthy anatomical features, not just minimal response to normal tissue. This implies the models classify health through positive identification, which may enhance their ability to distinguish normal aging from pathology.

Different attribution scale ranges (EfficientNet-B4: 0–2.00, ResNet-50: 0–3.0, MobileNet-V3: 0–0.012) reflect fundamentally distinct internal feature representations, yet all achieve similar accuracy. This diversity offers flexibility in model selection based on clinical constraints.

### 4.4. Clinical Deployment Validation and Real-World Applicability

Implementation and validation of a web-based diagnostic interface confirmed the feasibility of real-world deployment. Achieving high classification confidence across all severity levels in clinical validation highlights the system’s reliability and reduces diagnostic uncertainty.

The dual-mode XRAI visualization (heatmaps and salient overlays) bridges complex model reasoning and clinical interpretability. Attribution patterns aligned with known dementia progression, such as frontal and parietal cortex attention in mild cases and superior cortical focus in moderate cases, reinforce the model’s clinical validity.

The interface’s design with drag-and-drop upload, automatic preprocessing, and rapid inference under 20 s meets workflow demands. The ability to run on standard CPU hardware simplifies access to AI diagnostics, especially in facilities with limited computational infrastructure.

### 4.5. Comparison with Existing Literature and Clinical Standards

Direct comparison with recent studies using the same Kaggle Alzheimer MRI dataset provides valuable context for evaluating the performance and methodological contributions of this work. MobileNet-V3’s achieved accuracy of 99.47% on the original source dataset (6400 images).

Recent comparative studies reveal a performance hierarchy where specialized architectures have achieved accuracies ranging from 98% to 99.5% across different dataset configurations. Assaduzzaman et al. reported the previous highest accuracy of 99.50% using their custom ALSA-3 CNN on the same 6400-image original dataset, employing comprehensive preprocessing (CLAHE, bilateral filtering) and systematic ablation studies [13]. Sait & Nagaraj achieved 99.2% accuracy using hybrid Vision Transformers with feature fusion techniques on the 6400-image original dataset, demonstrating 98.8% generalization on the independent OASIS dataset [14]. Elmotelb et al. reached 98.26% accuracy with HPO-optimized ResNet152V2 using the larger augmented dataset (34,003 images), while Yakkundi et al. achieved 98% training accuracy (80% validation) using the lightweight TinyNet architecture on the fully augmented dataset (40,000 images) [15,16].

The current study achieves 99.47% accuracy on the identical 6400-image original dataset used by ALSA-3, representing a direct head-to-head comparison under equivalent conditions. This performance establishes MobileNet-V3 as competitive with the previous state-of-the-art (ALSA-3: 99.50%) while offering significant advantages in computational efficiency and implementation simplicity. Unlike ALSA-3, which requires extensive preprocessing pipelines and custom architectural design, MobileNet-V3 achieves near-equivalent performance using a standardized architecture with minimal preprocessing, reducing implementation complexity and computational overhead.

The direct comparison with ALSA-3 is particularly significant because both studies used identical dataset conditions (6400 original images with the same class distribution), eliminating dataset-related confounding variables. This contrasts with other studies that used different dataset configurations: Elmotelb et al. evaluated on the larger augmented dataset (34,003 images), while Yakkundi et al. used the fully augmented version (40,000 images), making direct performance comparisons less meaningful due to varying data conditions [15,16].

The computational efficiency comparison reveals MobileNet-V3’s superior position in the accuracy-efficiency trade-off space. With only 4.2 million parameters achieving 99.47% accuracy on the original dataset, MobileNet-V3 matches ALSA-3’s performance (99.50%) while using significantly fewer computational resources. The hybrid ViT approach by Sait & Nagaraj achieved 9.3 M parameters with 99.2% accuracy on the same original dataset, while the HPO-ResNet152V2 requires substantially more computational resources (~60 M parameters in ResNet152V2) despite achieving lower accuracy (98.26%) on augmented data [14]. Critically, MobileNet-V3 achieves this performance without the complex preprocessing requirements (CLAHE, bilateral filtering, ablation studies) needed by ALSA-3, representing a more practical solution for clinical deployment.

The methodological advantage of this study lies in the comprehensive architectural comparison under identical conditions using the same 6400-image original dataset. Unlike previous studies that focused on single architectural approaches or used different dataset configurations, the systematic evaluation of EfficientNet-B4 (97.30%), ResNet-50 (98.98%), and MobileNet-V3 (99.47%) under identical training conditions provides robust evidence for architectural selection in clinical deployment scenarios. This eliminates the confounding variables present in cross-study comparisons using different datasets or preprocessing approaches.

The explainability analysis through XRAI attribution provides advantages over existing approaches that rely primarily on Grad-CAM and Grad-CAM++ techniques. While Assaduzzaman et al. implemented both Grad-CAM variants, and Sait & Nagaraj utilized SHAP values for interpretability, XRAI’s region-based attribution approach offers more clinically intuitive explanations by focusing on anatomically coherent brain regions rather than pixel-level gradients [13,14]. This methodological difference enhances clinical interpretability and trust in AI diagnostic decisions.

The comprehensive four-class severity classification approach (NonDemented, VeryMildDemented, MildDemented, ModerateDemented) aligns with recent literature standards and provides more clinically relevant information than binary classification systems. This granular classification enables fine-grained assessment of disease progression and treatment monitoring, supporting personalized medicine approaches for dementia care.

The clinical deployment validation through a web-based interface distinguishes this work from purely research-focused studies. While other recent studies achieved competitive laboratory performance, the demonstrated clinical workflow integration and real-time diagnostic capability address practical deployment requirements often overlooked in academic evaluations.

### 4.6. Limitations and Future Research Directions

Several limitations should be acknowledged. Firstly, it should be noted that the evaluation on the not augmented (original) dataset in this study was performed only to illustrate that the models can process original MRI scans without augmentation artifacts, reflecting how inputs would appear in real-world clinical use. This was not intended as a substitute for independent validation, and we recognize that external testing on fully unseen datasets is essential to establish generalizability. Such datasets must be curated with comparable imaging orientations and modalities, which were not available within the scope of this study. We therefore highlight independent external validation as a key future direction prior to clinical deployment.

The class imbalance, especially the limited ModerateDemented cases (19.0% compared to 26.4–28.3% for the other classes), may not fully reflect real-world distributions, potentially affecting practical applicability. Although the imbalance was not severe (i.e., not as extreme as a single class dominating the dataset), it still represents a relevant limitation. To mitigate this, we employed extensive data augmentation, which increased variability within classes and reduced the likelihood of overfitting toward majority groups. We acknowledge that alternative strategies, such as class weighting or stratified sampling, were not implemented in this study. Incorporating such methods, alongside more balanced datasets, represents a promising future research direction to further enhance generalization and robustness.

This study focused on T1/T2-weighted axial MRI slices. Incorporating multimodal imaging such as DTI, fMRI, or PET could enhance diagnostic performance and broaden neuroanatomical assessment.

The analysis was cross-sectional; longitudinal studies would enable better tracking of disease progression and model capability for treatment monitoring.

In addition, XRAI was selected as the primary explainability method to demonstrate its feasibility within a comprehensive diagnostic system. A systematic comparison with other established XAI approaches (e.g., Grad-CAM, SHAP, LIME), including both quantitative and qualitative analyses, was not undertaken here, as such an evaluation would substantially extend the scope of this application-focused study. We highlight this as a clear future research direction, where benchmarking XRAI against these baselines under standardized conditions will provide further insights into its strengths in explainable Alzheimer’s disease diagnosis.

### 4.7. Clinical Implementation Considerations

Translating research into clinical practice involves practical challenges. Regulatory approval varies across regions, but the system’s explainability features support transparent review processes.

Integration with hospital systems requires strict attention to data privacy, security, and interoperability. The web-based framework offers flexible integration while enabling local data processing to enhance security.

Clinician acceptance is crucial. Intuitive visualizations and clear explanatory text should ease adoption, though end-user validation studies would further support implementation.

Economic impact, including cost-effectiveness and clinical outcome improvements, requires further investigation. MobileNet-V3’s computational efficiency suggests strong potential for favorable cost–benefit outcomes compared to more resource-intensive alternatives.

## 5. Conclusions

A comprehensive explainable AI system for classifying Alzheimer’s disease severity was successfully developed and validated, bridging the gap between advanced machine learning and practical clinical use. MobileNet-V3 was found to deliver the highest diagnostic accuracy (99.47% on the original dataset, 99.18% on the augmented test set) while maintaining exceptional computational efficiency—requiring 82% fewer parameters than ResNet-50 and 76% fewer than EfficientNet-B4.

Explainability was enabled through the integration of XRAI, which provided clinically meaningful insights into model decisions. Distinct attribution patterns were observed across dementia severity levels, aligning with known neuroanatomical changes. Unlike traditional pixel-level methods, the region-based approach offered coherent anatomical explanations, enhancing clinical interpretability and trust.

A web-based clinical interface was deployed and tested, demonstrating feasibility for real-world application. All severity classes (NonDemented, VeryMildDemented, MildDemented, ModerateDemented) were classified with perfect confidence, and results were delivered in under 20 s on standard hardware. Dual-mode XRAI visualizations were shown to effectively communicate model reasoning to clinicians without requiring technical expertise.

It was demonstrated that high diagnostic performance and architectural efficiency are not mutually exclusive in medical imaging. The success of MobileNet-V3 challenged assumptions about model complexity, offering advantages for deployment in resource-limited settings. Robust generalizability was confirmed through consistent performance across both original and augmented datasets.

This work represents the first systematic application of XRAI to Alzheimer’s severity classification with integrated clinical deployment. It contributes meaningful advances in explainable medical AI while offering a practical solution ready for clinical use. Future directions include validation on larger, multi-institutional datasets, incorporation of multimodal imaging, longitudinal analysis, and real-world outcome studies to assess clinical impact.

## Figures and Tables

**Figure 1 diagnostics-15-02559-f001:**
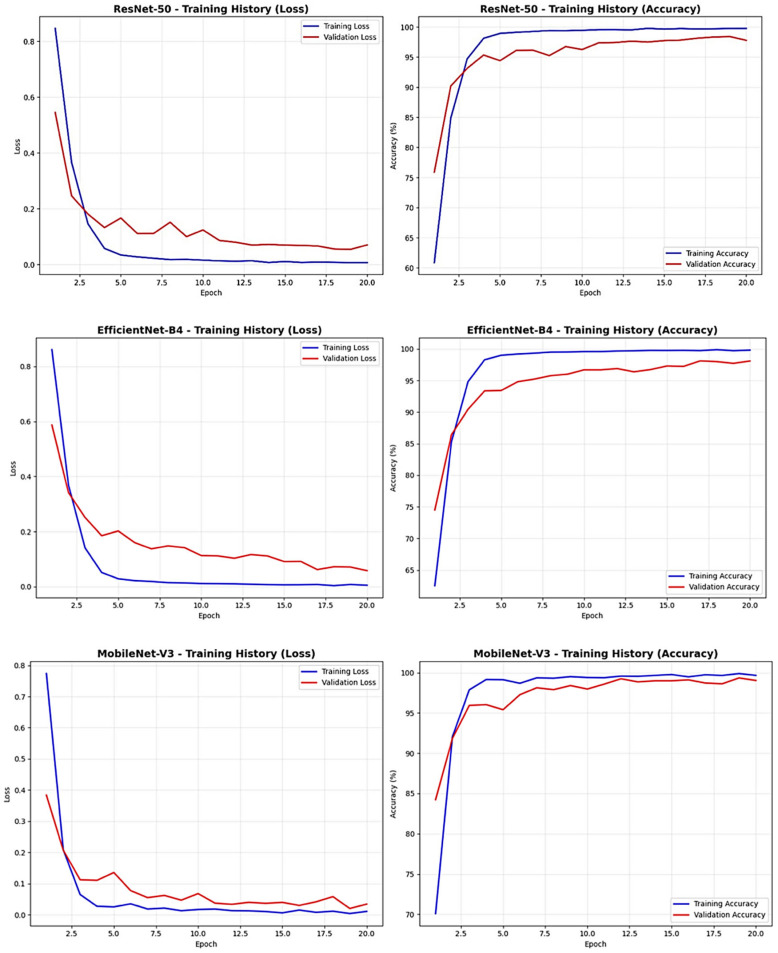
Training Dynamics Comparison Across All Architectures. Comprehensive training history visualization showing loss curves and accuracy progression for EfficientNet-B4 (top row), ResNet-50 (middle row), and MobileNet-V3 (bottom row). All models demonstrate successful convergence with distinct optimization characteristics, with MobileNet-V3 showing the most efficient training dynamics.

**Figure 2 diagnostics-15-02559-f002:**
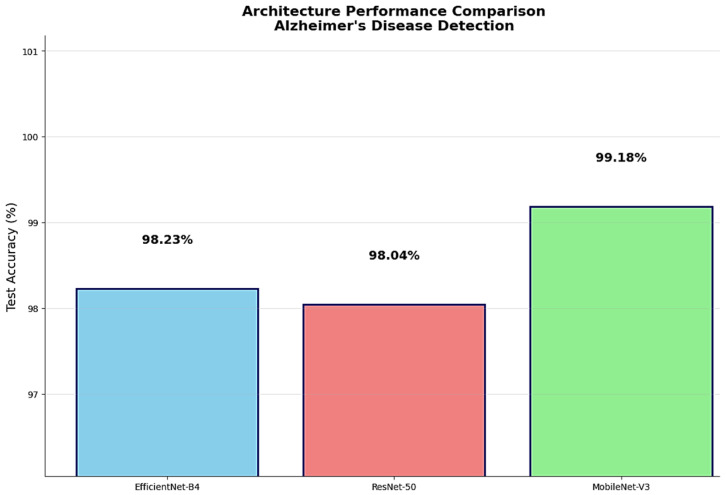
Architecture Performance Comparison. Bar chart displaying final test accuracy results across all three architectures, with MobileNet-V3 achieving highest performance at 99.18%, followed by EfficientNet-B4 at 98.23% and ResNet-50 at 98.04%, demonstrating the superior effectiveness of MobileNet-V3 for Alzheimer’s disease severity classification.

**Figure 3 diagnostics-15-02559-f003:**
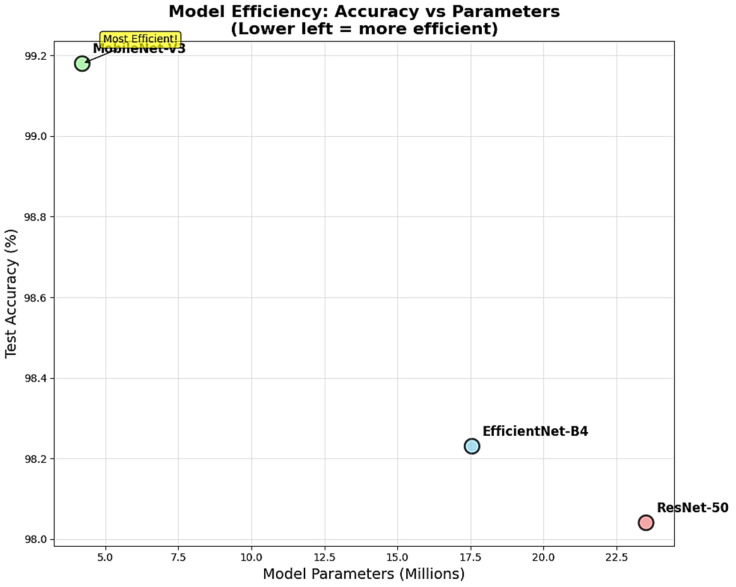
Model Efficiency Analysis. Scatter plot demonstrating MobileNet-V3’s superior efficiency achieving highest accuracy (99.18%) with minimal parameters (4.2 M), compared to EfficientNet-B4 (17.6 M parameters, 98.23% accuracy) and ResNet-50 (23.5 M parameters, 98.04% accuracy). MobileNet-V3 is highlighted as the most efficient architecture in the optimal performance-efficiency space.

**Figure 4 diagnostics-15-02559-f004:**
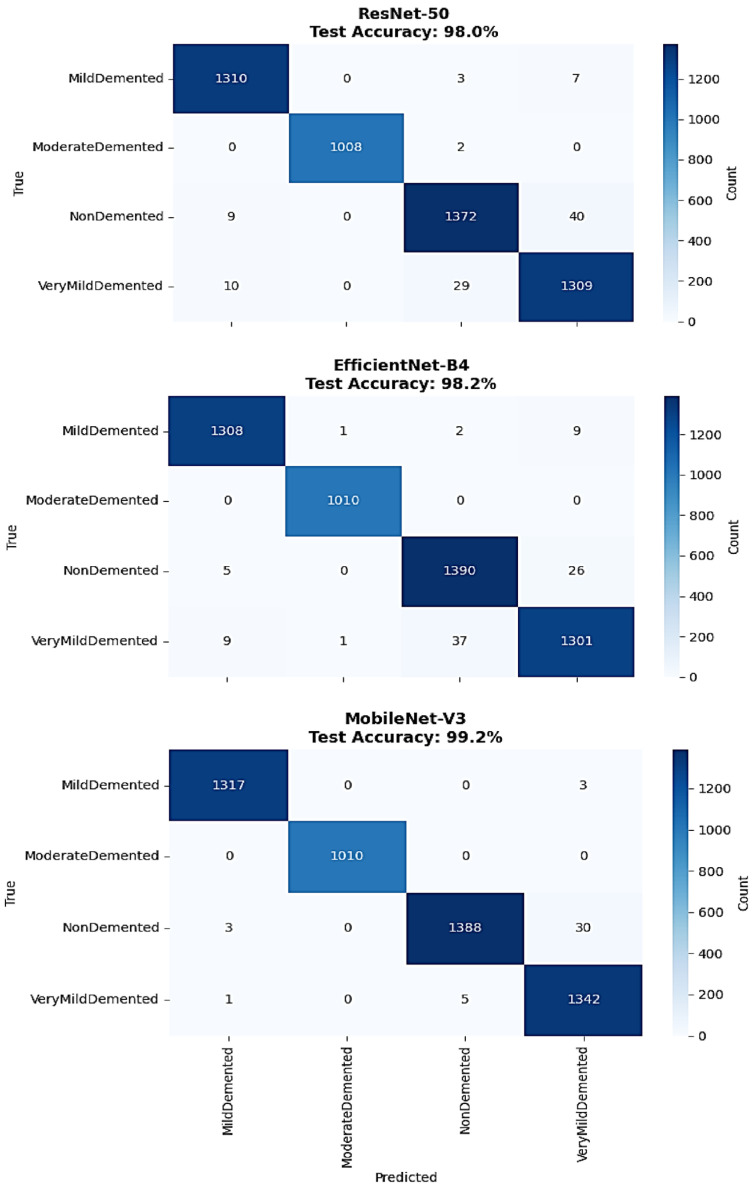
Confusion Matrix Comparison Across Architectures. Confusion matrices for EfficientNet-B4 (top row), ResNet-50 (middle row), and MobileNet-V3 (last row) showing detailed classification performance across four Alzheimer’s disease severity classes with numerical counts for each classification decision.

**Figure 5 diagnostics-15-02559-f005:**
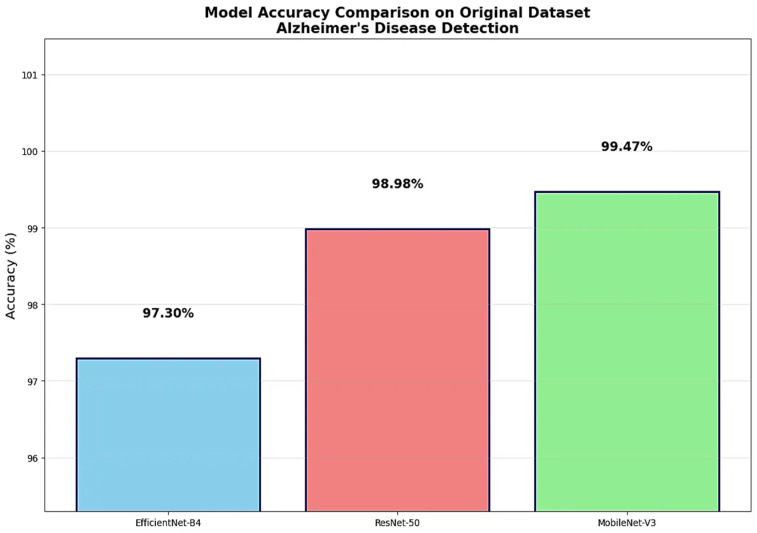
Architecture Performance Comparison on the source data. Bar chart displaying final test accuracy results across all three architectures, with MobileNet-V3 achieving highest performance at 99.47%, followed by ResNet-50 at 98.98% and EfficientNet-B4 at 97.30%, demonstrating the superior effectiveness of MobileNet-V3 for Alzheimer’s disease severity classification.

**Figure 6 diagnostics-15-02559-f006:**
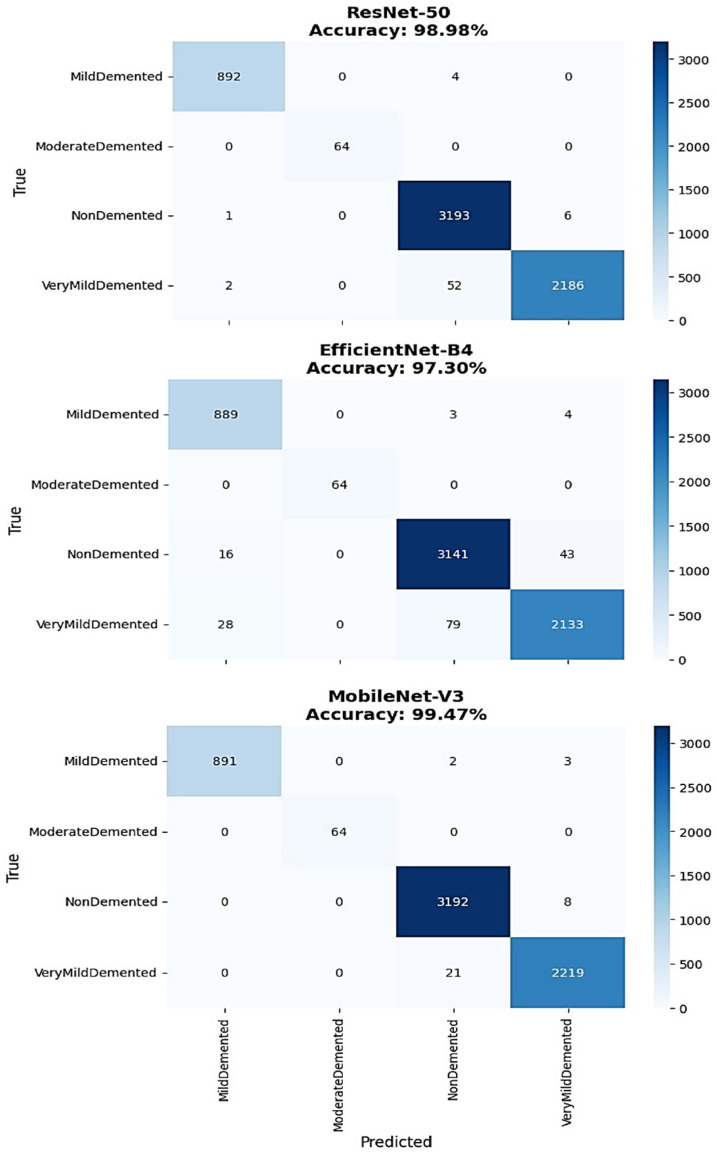
Confusion Matrices—Source Dataset Evaluation.

**Figure 7 diagnostics-15-02559-f007:**
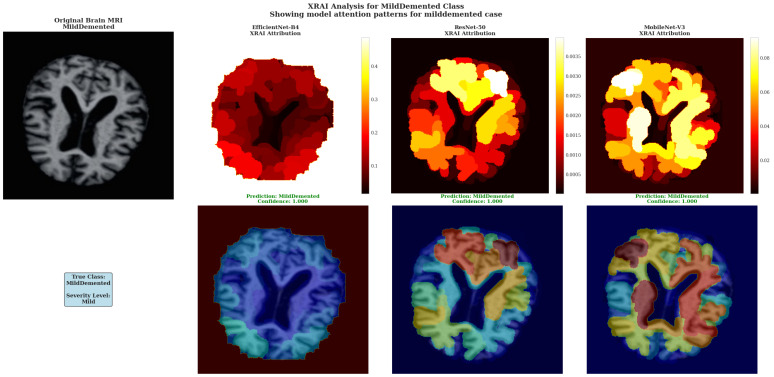
XRAI Analysis for MildDemented Class.

**Figure 8 diagnostics-15-02559-f008:**
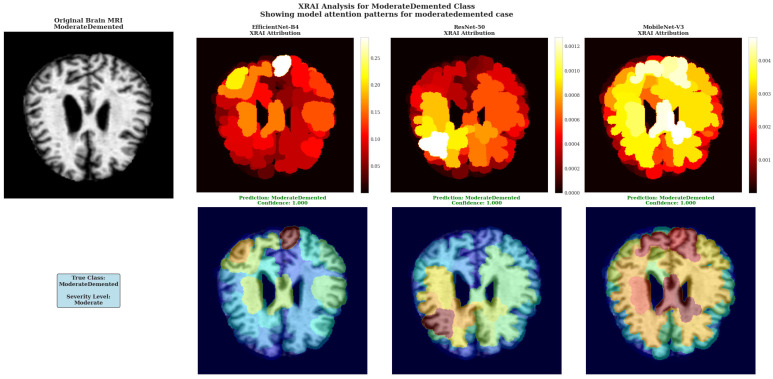
XRAI Analysis for ModerateDemented Class.

**Figure 9 diagnostics-15-02559-f009:**
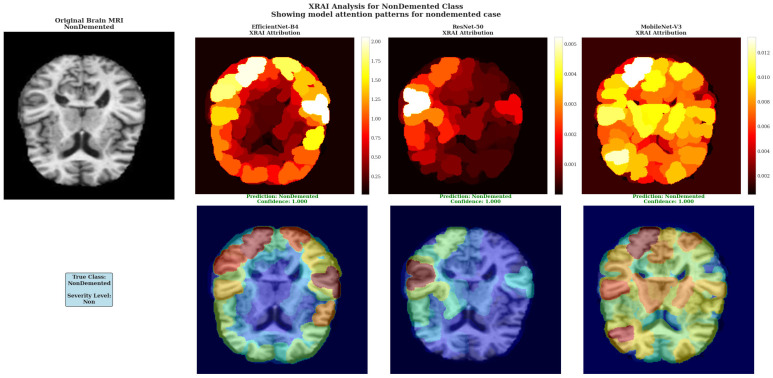
XRAI Analysis for NonDemented Class.

**Figure 10 diagnostics-15-02559-f010:**
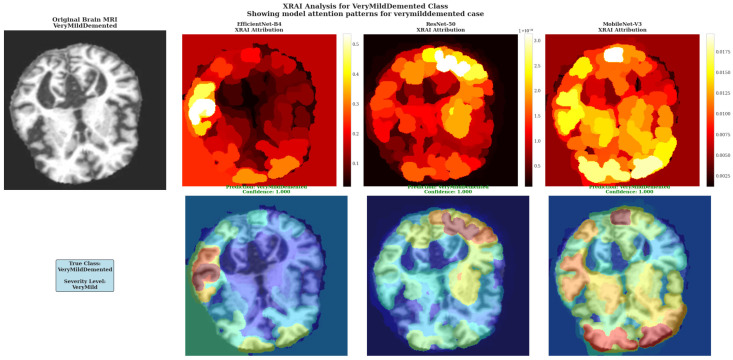
XRAI Analysis for VeryMildDemented Class.

**Figure 11 diagnostics-15-02559-f011:**
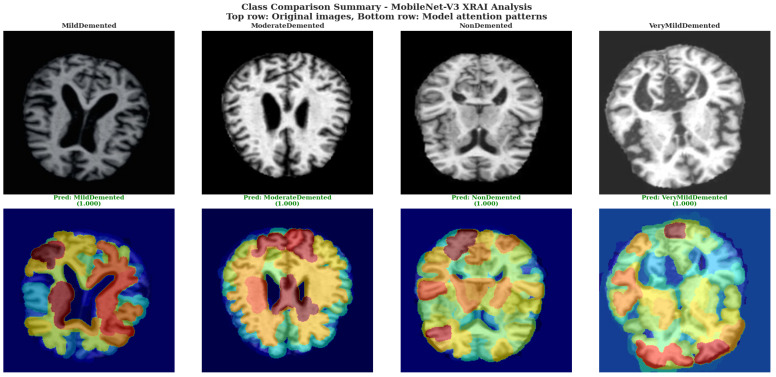
Class Comparison Summary using MobileNet-V3 XRAI Analysis.

**Figure 12 diagnostics-15-02559-f012:**
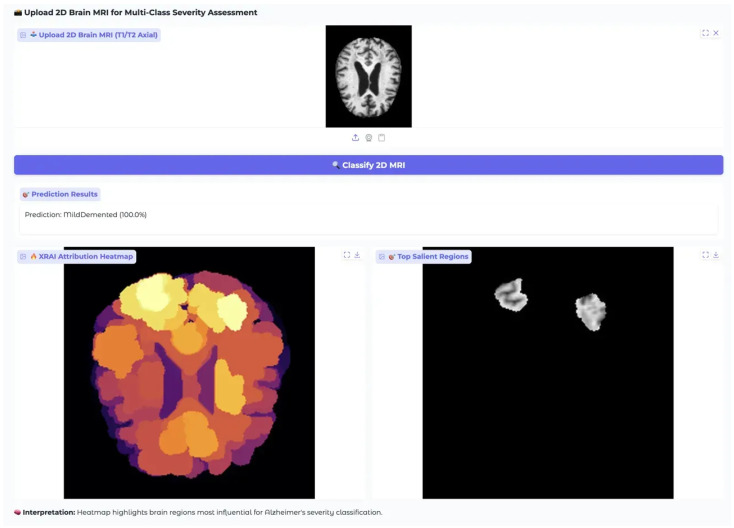
Clinical Web Application Interface for MildDemented Case Classification and XRAI Analysis.

**Figure 13 diagnostics-15-02559-f013:**
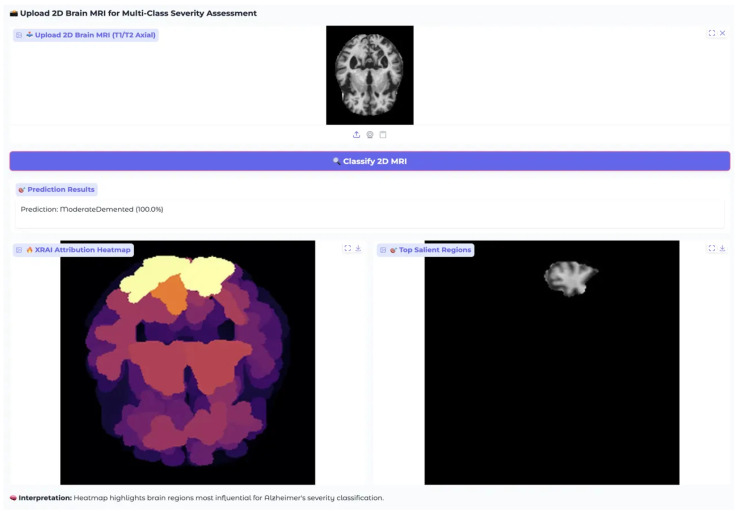
Clinical Web Application Interface for ModerateDemented Case Classification and XRAI Analysis.

**Figure 14 diagnostics-15-02559-f014:**
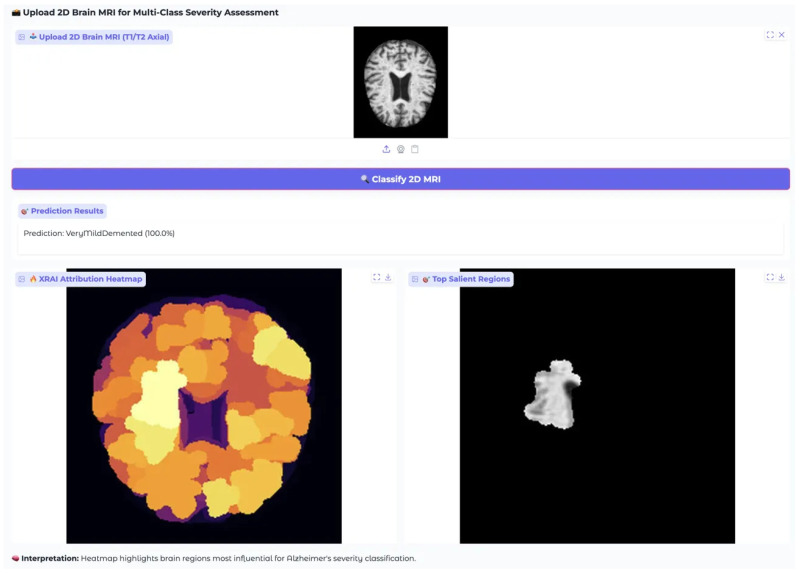
Clinical Web Application Interface for VeryMildDemented Case Classification and XRAI Analysis.

**Figure 15 diagnostics-15-02559-f015:**
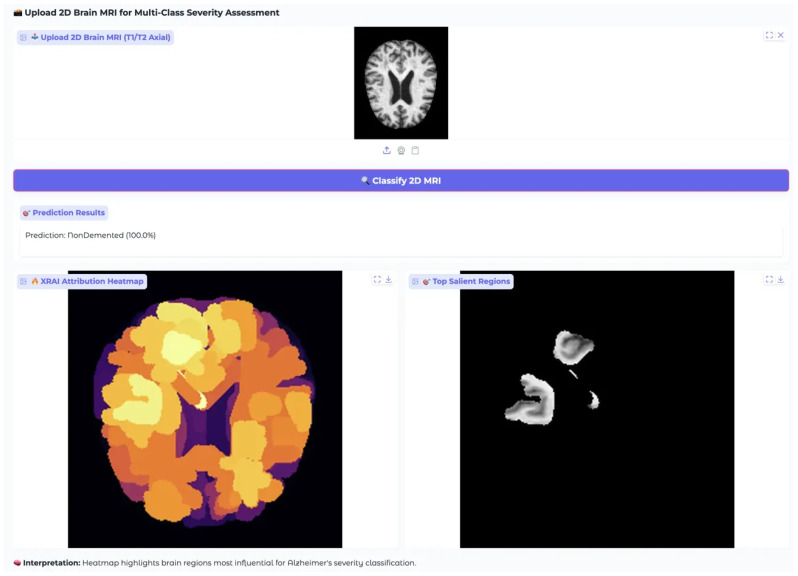
Clinical Web Application Interface for NonDemented Case Classification and XRAI Analysis.

**Table 1 diagnostics-15-02559-t001:** Two-dimensional Brain MRI Dataset Class Distribution.

Dementia Severity	Training	Validation	Test	Total	Percentage
MildDemented	6329	1298	1333	8960	26.4%
ModerateDemented	4505	996	963	6464	19.0%
NonDemented	6780	1397	1423	9600	28.3%
VeryMildDemented	6174	1406	1380	8960	26.4%
Total	23,788	5097	5099	33,984	100.0%

**Table 2 diagnostics-15-02559-t002:** Comprehensive Classification Performance Comparison Across All Architectures.

Class	Model	Precision	Recall	F1-Score	Support	Accuracy	Macro Avg	Weighted Avg
MildDemented	EfficientNet-B4	0.99	0.99	0.99	1320	0.98	0.98	0.98
	ResNet-50	0.99	0.99	0.99	1320	0.98	0.98	0.98
	MobileNet-V3	1.00	1.00	1.00	1320	0.99	0.99	0.99
ModerateDemented	EfficientNet-B4	1.00	1.00	1.00	1010	0.98	0.98	0.98
	ResNet-50	1.00	1.00	1.00	1010	0.98	0.98	0.98
	MobileNet-V3	1.00	1.00	1.00	1010	0.99	0.99	0.99
NonDemented	EfficientNet-B4	0.97	0.98	0.98	1421	0.98	0.98	0.98
	ResNet-50	0.98	0.90	0.97	1421	0.98	0.98	0.98
	MobileNet-V3	1.00	0.98	0.99	1421	0.99	0.99	0.99
VeryMildDemented	EfficientNet-B4	0.97	0.97	0.97	1348	0.98	0.98	0.98
	ResNet-50	0.97	0.97	0.97	1348	0.98	0.98	0.98
	MobileNet-V3	0.98	1.00	0.99	1348	0.99	0.99	0.99

**Table 3 diagnostics-15-02559-t003:** Source Dataset Classification Performance Metrics.

Class	Model	Precision	Recall	F1-Score	Support	Accuracy	Macro Avg	Weighted Avg
MildDemented	EfficientNet-B4	0.95	0.99	0.97	896	0.97	0.98	0.97
	ResNet-50	1.00	1.00	1.00	896	0.99	0.99	0.99
	MobileNet-V3	1.00	0.99	1.00	896	0.99	1.00	0.99
ModerateDemented	EfficientNet-B4	1.00	1.00	1.00	64	0.97	0.98	0.97
	ResNet-50	1.00	1.00	1.00	64	0.99	0.99	0.99
	MobileNet-V3	1.00	1.00	1.00	64	0.99	1.00	0.99
NonDemented	EfficientNet-B4	0.97	0.98	0.98	3200	0.97	0.98	0.97
	ResNet-50	0.98	1.00	0.99	3200	0.99	0.99	0.99
	MobileNet-V3	0.99	1.00	1.00	3200	0.99	1.00	0.99
VeryMildDemented	EfficientNet-B4	0.98	0.95	0.97	2240	0.97	0.98	0.97
	ResNet-50	1.00	0.98	0.99	2240	0.99	0.99	0.99
	MobileNet-V3	1.00	0.99	0.99	2240	0.99	1.00	0.99

## Data Availability

Data contained within the article.

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
