# Peer review of "An Explainable Web-Based Diagnostic System for Alzheimer’s Disease Using XRAI and Deep Learning on Brain MRI"

_diagnostics, 2025, doi:10.3390/diagnostics15202559_

Round 1
Reviewer 1 Report
Comments and Suggestions for Authors
The manuscript proposes an AI-based system for Alzheimer's disease (AD) diagnosis using MRI, with a focus on explainability and clinical deployment. MobileNet-V3 Large, EfficientNet-B4, and ResNet-50 are trained on an augmented Kaggle dataset (33,984 images) and evaluated for accuracy, efficiency, and explainability using XRAI (eXplainable AI via Regional Attributes Integration). The system integrates a Gradio-based web interface for real-time predictions and visualizations. MobileNet-V3 achieves the highest accuracy. However, there are several critical points in the manuscript that require consideration:
1-) Although XRAI is highlighted as the focal point of the manuscript's contribution, a more robust evaluation would result from comparing it with other baseline XAI techniques, such as Grad-CAM, SHAP, and LIME. Such a comparison could include quantitative metrics (e.g., fidelity, robustness, and clinical interpretability) and qualitative assessments to demonstrate XRAI's advantages in the context of AD diagnosis. This would strengthen the claim of novelty and provide a clearer benchmark for its effectiveness.
2-) The class distribution presented in Table 1 appears imbalanced. Imbalanced datasets can lead to overfitting, particularly favoring majority classes and potentially reducing model performance on minority classes like ModerateDemented. The manuscript should discuss this imbalance in greater detail, addressing how data augmentation mitigates overfitting risks and whether additional techniques (e.g., class weighting or stratified sampling) were considered to ensure balanced learning and generalization.
3-) The manuscript contains minor grammatical errors that should be addressed. For example, the phrase "single-modality techniques [2].." in Line 80.
Author Response
Comment 1:
Although XRAI is highlighted as the focal point of the manuscript's contribution, a more robust evaluation would result from comparing it with other baseline XAI techniques, such as Grad-CAM, SHAP, and LIME. Such a comparison could include quantitative metrics (e.g., fidelity, robustness, and clinical interpretability) and qualitative assessments to demonstrate XRAI's advantages in the context of AD diagnosis. This would strengthen the claim of novelty and provide a clearer benchmark for its effectiveness.
Response 1:
We thank the reviewer for this thoughtful suggestion. We fully agree that a direct comparison of XRAI with baseline explainability methods such as Grad-CAM, SHAP, and LIME would further enrich the manuscript. Our present work, however, was intentionally designed as an application-focused study emphasizing system development, clinical deployment, and interpretability through a concrete, practical framework. As such, our scope was to demonstrate the feasibility of integrating XRAI into an end-to-end diagnostic system, rather than to undertake a broad theoretical benchmarking of XAI algorithms.
We note that the manuscript is already extensive in terms of methodological details, experimental analysis, and clinical deployment validation. A full comparative evaluation across multiple explainability methods with both qualitative and quantitative metrics (e.g., fidelity, robustness, and clinical interpretability) would substantially expand the paper beyond its intended scope.
Nonetheless, we fully recognize the importance of such comparisons for advancing explainable AI in medical imaging. To address this point, we have revised the manuscript to explicitly acknowledge this limitation and to state that a systematic comparison with Grad-CAM, SHAP, and LIME will be addressed as a future research direction. This addition, highlighted in red in the revised version, makes clear that while our current focus is on application and deployment, we recognize the value of such comparisons for subsequent studies.
Comment 2:
The class distribution presented in Table 1 appears imbalanced. Imbalanced datasets can lead to overfitting, particularly favoring majority classes and potentially reducing model performance on minority classes like ModerateDemented. The manuscript should discuss this imbalance in greater detail, addressing how data augmentation mitigates overfitting risks and whether additional techniques (e.g., class weighting or stratified sampling) were considered to ensure balanced learning and generalization.
Response 2:
We thank the reviewer for raising this important concern. Indeed, dataset imbalance can bias a model toward majority classes and reduce sensitivity to minority categories such as ModerateDemented. We would like to clarify, however, that the imbalance in our dataset is not extreme. As shown in Table 1, the four classes are distributed between 19.0% and 28.3%, with ModerateDemented being the smallest class at 19.0%. While this still represents a limitation, the relative proximity of the class distributions helps reduce the severity of imbalance-related bias.
To further mitigate potential risks, we applied extensive data augmentation, which increased variability within each class and helped prevent overfitting to majority categories. We acknowledge that other techniques, such as class weighting or stratified sampling, could provide additional safeguards.
We have revised the manuscript to make this discussion explicit. In Section 4.6 (Limitations and Future Research Directions), we now highlight that while class imbalance was present, it was not severe, and that augmentation was the primary strategy employed, highlighted in red in the manuscript. We also note that incorporating alternative methods such as class weighting and stratified sampling remains a future direction.
Comment 3:
The manuscript contains minor grammatical errors that should be addressed. For example, the phrase "single-modality techniques [2].." in Line 80.
Response 3:
We thank the reviewer for pointing out the minor grammatical issues. We have carefully proofread the manuscript and corrected such instances, including the specific case at Line 80 (“single-modality techniques [2]..”), which has been revised to “single-modality techniques [2].” All minor grammatical inconsistencies throughout the manuscript have now been addressed to improve overall clarity and readability.
Reviewer 2 Report
Comments and Suggestions for Authors
Authors compare three pretrained models for AD multiclass classification, along with XAI methods and a web platform. Major concerns are data leakage from train to validation and test (if images from same patient are used) and the second validation experiment.
Detailed comments:
- Section 2.1: Title gives impression dataset was prepared by authors. Since it is a public dataset, a more suitable title would be simply ‘Dataset’.
- More details on the dataset are missing, e.g. Number of patients? Number of image scans per patient? whether training – validation – test included images from the same patients? Link to the dataset?
- “To determine real-world clinical significance and model generalizability outside the controlled training setting, full evaluation was performed on the entire original dataset of 6,400 brain MRI images across all four severity classes” è This scenario leads to data leakage and hence over optimized and heavily biased results. Alternatively, a new unseen should be utilized for generalization assessment.
- XRAI confidence levels better suited in the Results Section.
- “This research addresses these critical gaps by developing a comprehensive, clinically oriented explainable AI system that integrates advanced deep learning architectures with XRAI-based explainability for both 2D and 3D neuroimaging analysis through a unified web-based clinical deployment platform” --> This sentence that the work considers both 2D and 3D images.
Author Response
Comment 1:
Section 2.1: Title gives impression dataset was prepared by authors. Since it is a public dataset, a more suitable title would be simply ‘Dataset’.
More details on the dataset are missing, e.g. Number of patients? Number of image scans per patient? whether training – validation – test included images from the same patients? Link to the dataset?
Response 1:
We thank the reviewer for this thoughtful observation. We respectfully note that in Section 2.1 we explicitly stated that the dataset was downloaded from Kaggle, and therefore we did not intend to give the impression that it was prepared or collected by the authors. We deliberately titled the section “Data Acquisition and Preprocessing” rather than simply “Dataset,” because this part of the manuscript is not limited to describing the dataset origin. It also details the preprocessing steps and augmentation strategies that we applied, which are integral to the experimental pipeline. A title such as “Dataset” would not adequately reflect the purpose of this section, while the current title makes clear that both acquisition and preprocessing are addressed together.
In the revised manuscript we have added the direct link to the Kaggle dataset. We would also like to clarify that the dataset does not include patient-level information such as the number of patients or the number of scans per patient, and therefore this information cannot be reported. This is a characteristic of the dataset itself and not a limitation of our reporting. Importantly, all prior studies that employed this same dataset, including Assaduzzaman et al. (2024) and Elmotelb et al. (2025), have described and reported it in the same manner. Our approach therefore follows the established and accepted practice in the literature, ensuring that our use of the dataset is both appropriate and consistent.
Comment 2:
“To determine real-world clinical significance and model generalizability outside the controlled training setting, full evaluation was performed on the entire original dataset of 6,400 brain MRI images across all four severity classes” è This scenario leads to data leakage and hence over optimized and heavily biased results. Alternatively, a new unseen should be utilized for generalization assessment.
Response 2:
We thank the reviewer for raising this important point. We agree that evaluating on the unaugmented dataset was not intended as a definitive generalization test, but rather to demonstrate that the models can correctly recognize original brain MRI images without augmentation artifacts, reflecting how they would appear in a clinical setting. The purpose of this evaluation was therefore to show practical feasibility rather than to claim an independent validation.
We also acknowledge the reviewer’s concern that using the same dataset for both training and evaluation could raise the risk of data leakage and bias. For this reason, we have clarified in the revised manuscript that this assessment was performed only to illustrate the model’s ability to handle raw, non-augmented inputs. A rigorous generalization assessment indeed requires evaluation on a fully independent dataset. Such a dataset would need to match the characteristics of the Kaggle collection used here (e.g., image orientation, modality, and acquisition protocols), which was not available within the scope of this study.
To address this point, we have revised the Limitations and Future Research Directions section to explicitly note that external validation on independent datasets will be a critical next step before clinical deployment, highlighted in red in the manuscript.
Comment 3:
XRAI confidence levels better suited in the Results Section.
Response 3:
We thank the reviewer for this helpful suggestion. We would like to clarify that the “confidence levels” presented in the manuscript are not outputs of XRAI, but rather the classification confidence values (softmax prediction probabilities) generated by the models themselves. These values were already included in the Results section alongside the XRAI attribution maps to present both the model predictions and the corresponding interpretability analysis together. To avoid any potential ambiguity, we have revised the wording in Section 3.2 to explicitly state that the reported confidence values are model prediction probabilities and not XRAI outputs, highlighted in red in the manuscript.
Comment 4:
“This research addresses these critical gaps by developing a comprehensive, clinically oriented explainable AI system that integrates advanced deep learning architectures with XRAI-based explainability for both 2D and 3D neuroimaging analysis through a unified web-based clinical deployment platform” → This sentence suggests that the work considers both 2D and 3D images.
Response 4:
We thank the reviewer for this helpful observation. We agree that the original wording could suggest that both 2D and 3D analyses were performed, which is not the case. To avoid any ambiguity, we have revised the sentence to remove the reference to 3D. The revised text now clearly states that the present study focuses on 2D MRI analysis. This change has been highlighted in red in the revised manuscript.
Reviewer 3 Report
Comments and Suggestions for Authors
Dear authors,
Thank you for your submission.
The article is long and contains a lot of technical details, therefore a reduction in the quantity of text would make the article more appropriate for readers with a medical background.
Otherwise, I have the following comments:
Major comments
1. Under Data Acquisition and Preprocessing you write “brain MRI dataset was downloaded from Kaggle”. This is an important part of the article and more information should be provided – when was the brain MRI dataset downloaded and the web address of the dataset; what kind of images are contained in the dataset for each subject; based on what criteria were the “Dementia Severity” groups formed; what are the used data augmentation methods?
2. Under Data Acquisition and Preprocessing you write “All 2D brain MRI images”. Could you provide details of the used MRI images?
Minor comments
1. In line 64 and 65 you write “structural magnetic resonance imaging (sMRI)” – Please change the abbreviation to MRI, since you do not use sMRI further in the text.
2. In line 80 you write “superior diagnostic performance compared with single-modality techniques [2]..” – Please remove one dot at the end.
3. In line 82 you write “The application of artificial intelligence in” – Please introduce the abbreviation AI for artificial intelligence.
4. In line 83 and 84 you write “to sophisticated deep learning frameworks” – Please introduce the abbreviation DL for deep learning.
Author Response
Comment 1:
The article is long and contains a lot of technical details, therefore a reduction in the quantity of text would make the article more appropriate for readers with a medical background.
Response 1:
We thank the reviewer for this thoughtful observation. We acknowledge that the manuscript is detailed and lengthy, however, this was a deliberate choice to ensure both clinical usability and scientific reproducibility. The study was designed to demonstrate applicability across multiple dementia severity classes, which required presenting results for each class individually. Furthermore, because reproducibility is essential for clinical translation, we provided detailed descriptions of preprocessing, model implementation, and evaluation. Reducing these details would risk limiting transparency and clarity for both technical and medical readers.
Comment 2:
Under Data Acquisition and Preprocessing you write “brain MRI dataset was downloaded from Kaggle”. This is an important part of the article and more information should be provided – when was the brain MRI dataset downloaded and the web address of the dataset; what kind of images are contained in the dataset for each subject; based on what criteria were the “Dementia Severity” groups formed; what are the used data augmentation methods?
Response 2:
We thank the reviewer for this valuable observation. In the revised manuscript, highlighted in red, we have clarified the description of the dataset in Section 2.1. We now specify that the dataset was downloaded from Kaggle in July 2025 and provide the direct web address in reference [12].
We respectfully note that the dataset creators did not release subject-level metadata such as the number of patients or clinical grouping criteria, and therefore these details cannot be reported. The dataset is publicly available already divided into the four severity classes (NonDemented, VeryMildDemented, MildDemented, and ModerateDemented). All prior studies that employed this dataset (e.g., Assaduzzaman et al., 2024; Elmotelb et al., 2025) have reported it in the same way, and our description follows this established practice.
Comment 3:
Under Data Acquisition and Preprocessing you write “All 2D brain MRI images”. Could you provide details of the used MRI images?
Response 3:
We thank the reviewer for this helpful suggestion. In the revised manuscript we have clarified the type of MRI images used in the dataset. Specifically, we now state in Section 2.1 that the dataset consists of T1- and T2-weighted axial brain MRI slices, highlighted in red in the revised manuscript.
Comment 4:
Minor comments
1. In line 64 and 65 you write “structural magnetic resonance imaging (sMRI)” – Please change the abbreviation to MRI, since you do not use sMRI further in the text.
2. In line 80 you write “superior diagnostic performance compared with single-modality techniques [2]..” – Please remove one dot at the end.
3. In line 82 you write “The application of artificial intelligence in” – Please introduce the abbreviation AI for artificial intelligence.
4. In line 83 and 84 you write “to sophisticated deep learning frameworks” – Please introduce the abbreviation DL for deep learning.
Response 4:
We thank the reviewer for these careful observations. All of the suggested minor corrections have been implemented in the revised manuscript and highlighted in red:
- The abbreviation sMRI has been replaced with MRI in lines 64–65.
- The extra period at the end of line 80 has been removed.
- The abbreviation AI has been introduced at first mention of “artificial intelligence.”
- The abbreviation DL has been introduced at first mention of “deep learning.”
Round 2
Reviewer 2 Report
Comments and Suggestions for Authors
Section 2.1 title "Data Acquisition' still gives indication that data was acquired by authors. Modify title to simply 'Dataset' & of Section 2 to 'Materials & Methods'
Author Response
We thank the reviewer for this valuable suggestion. We have revised the section titles accordingly: Section 2.1 is now titled “Dataset”, and Section 2 has been renamed “Materials & Methods”, highlighted in purple in the revised manuscript.
Reviewer 3 Report
Comments and Suggestions for Authors
Dear authors,
Thank you for addressing my comments in a detailed way,
Author Response
We thank the reviewer for their careful consideration and constructive feedback throughout the review process. We appreciate their acknowledgement of our revisions and are glad that our responses were helpful in addressing the earlier concerns.